# Sound Verification of Deployed Neural Networks

## Abstract

Verification methods aim at mathematically proving desirable properties of neural networks, such as robustness to adversarial perturbations. A verifier is sound if and only if it never claims that a neural network has the desired property when it does not. It was shown recently that none of the currently known verifiers that are claimed to be sound are guaranteed to be sound when considering the deployed version of the verified network. Due to this, all the known verifiers are vulnerable to certain backdoor attacks, where an adversarial network passes verification but, in reality, it exhibits adversarial behavior in specific deployment environments. So far, it has been suspected that sound verification is prohibitively expensive if we wish to verify all possible executions—including parallel and stochastic ones—in deployment. *We are the first to propose an efficient error bounding technique that most known verifiers can apply to become practically sound.* The technique enables both interval bound propagation and symbolic propagation methods to remain sound even if the deployment environment randomly selects a valid ordering and parenthesizing of the arithmetic operations to compute the network. We present a theoretical foundation for our approach and demonstrate empirically that our technique indeed discovers all known deployment-specific attacks, introducing only a limited performance overhead.

## 1 Introduction

The goal of the formal verification of neural networks is to provide mathematical guarantees for certain properties, such as robustness to adversarial perturbations. Numerous studies (Szász et al. (2025); Zombori et al. (2021); Jia & Rinard (2021)) have shown that existing verifiers focus on providing safety guarantees for the *theoretical model* of the network, while overlooking some of the practical aspects of floating-point environments (Schlögl et al. (2023); Villa et al. (2009)) resulting in *incorrect verification results in practice*.

Szász et al. (2025) make the strongest claim, namely that none of the known verifiers are sound in practical deployment. This is due to the fact that the theoretical and practical networks are different mathematical objects, and the target of verification should be the practical network. In the simplest case of computing a sum, the theoretical computation is a well-defined operation with a unique result, while the practical computation depends on the ordering and parenthesizing of the addition operations—in other words, the binary *expression tree* that is used to compute the sum—due to the non-associativity of floating-point arithmetic. However, the exact expression tree is normally not known in advance in deployment (Schlögl et al. (2023); Villa et al. (2009)).

Furthermore, Szász et al. (2025) argue that this deficiency of current verification algorithms opens the door to practically viable backdoor attacks that are activated by certain targeted environments, while the network with the backdoor passes verification.

Here, we offer a solution to this problem. We bound the backward error of inner products building on a method proposed by Higham (2002), thereby defining a real-valued interval that includes the floating-point result of every possible expression tree for the inner product in deployment. We then generalize this technique to ReLU networks.

While our bounding technique could be incorporated into most verifiers, here, we implement two simple verification algorithms based on our results: one using interval propagation, and the other

based on symbolic propagation. We prove that both verifiers are practically sound, that is, they never label a network robust when there is at least one possible expression tree that violates robustness. While we allow any expression tree to be selected dynamically, we assume that no approximate algorithms (e.g., approximate matrix multiplication) are used in the deployment environment and no under- or overflow occurs.

We also demonstrate in practice that our new verifiers are able to verify a diverse set of adversarial networks that fool existing verifiers. In addition, our bounding technique has a small overhead in terms of both runtime and output range approximation when verifying benign networks.

We summarize our main contributions below.

- We propose a bounding technique that allows us to include the floating-point result of every possible expression tree that computes a ReLU network in deployment.
- We demonstrate the technique in the context of both interval bound propagation and symbolic propagation by proposing simple algorithms and proving their correctness.
- We demonstrate empirically that our novel algorithms are able to correctly verify a diverse set of adversarial networks from the literature, and that the bounding technique incurs an overhead of only about 25% and 4% extra runtime for interval propagation and symbolic propagation, respectively, over the task we examined.

## 2 RELATED WORK

Due to extensive research in the field of neural network safety, a wide range of verifiers have been developed so far. Good overviews of the topic are offered by Li et al. (2023); Albarghouthi (2021); Huang et al. (2020); Liu et al. (2021). These verifiers can be classified in numerous ways, but here, we will focus on sound verifiers. A verification algorithm is called *sound* if every property it proves to be true is indeed true, and *complete* if it successfully proves every true property.

Since neural network verification that is both sound and complete is known to be NP-complete (Katz et al. (2017)), most verifiers aim to over-approximate the behavior of the network, thereby achieving soundness but sacrificing completeness. These algorithms typically use interval propagation or symbolic propagation to compute bounds on the output. Interval propagation is based on interval arithmetic (Alefeld & Herzberger (1983)), often called interval bound propagation or IBP (Gowal et al. (2018); Xu et al. (2020)) in the context of neural network verification. Symbolic approaches propagate interval definitions symbolically, which allows for tighter bounds. Most symbolic methods are based on a linear over-approximation of the ReLU function. Some employ single-neuron approximations (Singh et al. (2018; 2019a); Zhang et al. (2018); Xu et al. (2021)), while others utilize multi-neuron approximations to achieve higher precision (Ferrari et al. (2022); Müller et al. (2022)).

Methods that are claimed to be sound and complete often formalize the verification problem as an SMT (Katz et al. (2017); Ehlers (2017)) or mixed-integer linear programming (MILP) model (Dutta et al. (2018); Tjeng et al. (2017)) and apply solvers to solve them. Some other algorithms are built upon the branch-and-bound (BaB) (Bunel et al. (2020)) framework. These algorithms apply fast sound algorithms during the bounding phase and then use different heuristics to perform the branching. Detailed explanations of these methods are offered by Wang et al. (2021); Palma et al. (2021); Ferrari et al. (2022); Xu et al. (2021).

In recent years, numerous studies have examined how deployment environments influence the behavior of numerical programs, including neural networks (Schlögl et al. (2023); Villa et al. (2009); Shanmugavelu et al. (2024); Cordeiro et al. (2025); Zhang et al. (2025); Möller et al. (2025)), as well as the reliability of verification algorithms (Zombori et al. (2021); Szász et al. (2025); Jia & Rinard (2021)). These studies demonstrated that the theoretical and practical soundness of a verification algorithm do not necessarily align. They constructed malicious networks that are reported to be safe by state-of-the-art sound verifiers but exhibit incorrect behavior in certain deployment environments.

Zombori et al. (2021) defined an adversarial network that exploits the numerical error introduced by LP solvers in LP/MILP model-based verification. Jia & Rinard (2021) proposed an adversarial attack on verifiers by finding adversarial (input, network) pairs that are incorrectly verified by MIPVerify.

Szász et al. (2025) went a step further, arguing that the theoretical and deployed (practical) models are different mathematical objects, and verification must consider the details of the deployment environment. The authors defined different adversarial networks and showed that none of the existing verifiers are able to detect their malicious behavior. They highlighted that the primary source of the vulnerability is that none of the available bounding algorithms can compute bounds that, regardless of the order of additions, contain all possible outputs.

Shanmugavelu et al. (2025) showed that—due to summation not being associative in floating-point—the correct classification of inputs near the decision boundary can strongly depend on the order of floating-point operations that would otherwise be associative in real arithmetic.

The impact of numerical representation has also been investigated in the context of quantization: recent works demonstrated that quantization errors can be exploited to introduce harmful behaviors into models that appear safe in higher precision (Egashira et al. (2024; 2025)).

We should also mention a different approach for circumventing the floating-point problem: using quantized neural networks that avoid rounding errors by using integer arithmetic (Henzinger et al. (2021); Huang et al. (2024); Zhang et al. (2023)).

Despite these research efforts, the problem of order-independent bounding of floating-point sums—and thus floating-point sound verification—remains, to the best of our knowledge, unresolved.

## 3 BACKGROUND AND NOTATIONS

Our notations are based on those of Szász et al. (2025) and Ferrari et al. (2022).

### 3.1 VERIFICATION PROBLEM

Let $f(.; \theta) : \mathbb{R}^n \mapsto \mathbb{R}^m$ be a neural network, where $\theta \in \mathbb{R}^k$ is a vector of real parameters. In this paper, we work with classifier neural networks, that is, $m$ denotes the number of output classes. The class label of an input $x$ is given by $y(x) = \arg\max_i f(x; \theta)_i$, that is, the index of the maximal output value.

In robustness verification, the central question is whether the classification result of a model remains unchanged when its input is perturbed within a bounded neighborhood of a given input point $x^*$. Let $D \subseteq \mathbb{R}^n$ denote this neighborhood, typically specified by a norm $\| \cdot \|_p$ and a perturbation radius $\epsilon$: $D_{\epsilon,p}(x^*) = \{x : \|x - x^*\|_p \leq \epsilon\}$. We define a safety property $P \subseteq \mathbb{R}^n$ that contains all inputs whose predicted class is identical to that of $x^*$. This property can be expressed as $P(x^*) = \{x : y(x) = y(x^*)\}$. The robustness requirement is then $D_{\epsilon,p}(x^*) \subseteq P(x^*)$. To make the verification problem more tractable, we apply a transformation to the network by appending an affine layer whose $i$-th output component is $f(x; \theta)_{y(x^*)} - f(x; \theta)_i$. Under this reformulation, the verification task becomes proving

$$\forall x \in D_{(\epsilon,p)}(x^*), \ f(x; \theta) \geq 0, \tag{1}$$

where $f$ now denotes the modified model. From now on, $f$ will refer to this transformed function.

### 3.2 FLOATING-POINT REPRESENTATION

A floating-point value can be expressed in the form $s \cdot b^e$, where $s$ is the signed significand (or mantissa), $b$ is the base (commonly $b = 2$), and $e$ is the exponent. Different floating-point formats primarily vary in the number of bits allocated to store the significand and the exponent. The IEEE 754-1985 standard introduced the widely used double-precision and single-precision formats. In double precision, also referred to as *binary64*, the exponent is stored using 11 bits, and the significand using 52 bits. In single precision (*binary32*), the exponent is allocated 8 bits, while the significand receives 23 bits. In both cases, one additional bit encodes the sign of the number.

Since floating-point arithmetic operates with finite precision, the results of operations are rounded. The most commonly used rounding mode is round-to-nearest, where the output is rounded to the nearest representable number. Rounding makes floating-point computations *order-dependent* (i.e., *non-associative*) and *precision-dependent*. The same expression may yield different results depend-

ing on the order of operations and the number of bits used to represent the significand. We now illustrate this two prominent problems through examples.

**Non-associative operations.** Changing the order of operations in a summation can produce different outputs. For example, $2^{53} + 1 - 2^{53}$ equals 0 in *binary64* arithmetic, when using this fixed order. However, reordering the operations as $2^{53} - 2^{53} + 1$ equals 1 under the same settings.

**Precision.** The number of bits in the significand varies between formats, affecting the result. For example, $2^{24} + 1 - 2^{24}$ evaluates to 0 in *binary32*, but to 1 in *binary64*.

### 3.3 VERIFICATION OF DEPLOYED MODELS

Here, we briefly summarize the concepts introduced by Szász et al. (2025), where it was emphasized that the deployed model and the theoretical model $f$ are different mathematical objects. Based on this observation, the verification problem should also be revisited in the deployed setting. The differences arise mainly from numerical representation, ordering of numerical operations, and stochastic behavior due to parallelization and hardware properties. The *deployed model* and the *deployed verification problem* account for these effects.

**Deployed Model.** Let $\mathcal{E}$ denote the deployment environment, encompassing all the characteristics that influence computation. Let $r(\cdot; \theta, \mathcal{E}) : X \to Y$ be the function actually computed when the theoretical model $f(\cdot; \theta)$ is deployed in $\mathcal{E}$. Here, $X \subset \mathbb{R}^n$ and $Y \subset 2^{\mathbb{R}^m}$ (that is, subsets of $\mathbb{R}^m$ due to stochasticity) represent the sets of inputs and outputs that are *representable* in $\mathcal{E}$. Even in deterministic environments, $r$ is not necessarily equivalent to $f$ due to numerical errors. In stochastic environments, such as those involving extensive parallelization or random execution order, $r(x; \theta, \mathcal{E})$ can produce multiple possible outputs. We therefore model $r$ as mapping into subsets of $\mathbb{R}^m$ containing all outputs with non-zero probability.

**Deployed Verification Problem.** For an input point $x^* \in X$, the deployed verification problem is

$$\forall x \in D_{\epsilon, p, \mathcal{E}}(x^*) \, \forall z \in r(x; \theta, \mathcal{E}), \; z \geq 0, \tag{2}$$

where $D_{p, \epsilon, \mathcal{E}}(x^*) = \{x \in X : \|x - x^*\|_p \leq \epsilon\}$. The safety property in this setting is $P_{\mathcal{E}}(x^*) = \{x \in X : \forall z \in r(x; \theta, \mathcal{E}), \; z_{y(x^*)} = \max_i z_i\}$, which in general satisfies neither $P_{\mathcal{E}} \subset P$ nor $P \subset P_{\mathcal{E}}$ (Szász et al. (2025)).

### 3.4 BOUNDING ALGORITHMS AND THEIR DEPLOYMENT ISSUES

Sound verification requires computing lower bounds of $f$ or $r$ over a domain, as in eqs. (1) and (2), perhaps accounting for numeric errors as well. Here, we detail the two most commonly used approaches: interval bound propagation and symbolic evaluation.

**Interval bound propagation (IBP)** executes each arithmetic operation using intervals. For example, the sum of the intervals $[l_1, u_1]$ and $[l_2, u_2]$ is $[l_1 + l_2, u_1 + u_2]$. Here, $[l_1 + l_2, u_1 + u_2]$ guarantees the bounding of all possible $s = a_1 + a_2$, where $a_1 \in [l_1, u_1]$ and $a_2 \in [l_2, u_2]$. Other operators like subtraction, multiplication, and division can also be extended to work on intervals. A big advantage of this method is that very efficient implementations are available that can exploit GPU tensor operations (Gowal et al. (2018); Zhang et al. (2020)).

**Problem with IBP.** When using IBP in deployment, outward rounding can be applied to guarantee the bounding of the real result. That is, the lower bound of the output interval is rounded towards $-\infty$ and the upper bound is rounded towards $+\infty$. For example, $[1, 1] + [2^{53}, 2^{53}]$ evaluates to $[2^{53}, 2^{53} + 2]$ in a *binary64* environment, successfully bounding the real-valued result $2^{53} + 1$. The next example illustrates a problem with IBP, though: the expression $([1, 1] + [1, 1]) + [2^{53}, 2^{53}]$ evaluates to $[2^{53} + 2, 2^{53} + 2]$ in *binary64*, which does not include the *binary64* result of the expression $(2^{53} + 1) + 1 = 2^{53}$. However, in deployment, it is uncertain which expression tree (ordering and parenthesizing) will be used, so **all expression trees should be covered**.

**Symbolic propagation methods** maintain symbolic expressions instead of numeric intervals. The input is typically represented as variables (interval or scalar), and during propagation, only the coefficients of the expressions are updated. Here, we will consider only *linear interval expressions*

developed by Miné (2004)

$$L(x) = [l_0, u_0] + \sum_{i=1}^{n} [l_i, u_i] x_i \tag{3}$$

that are the most common in neural network verification, as they are easy to represent numerically and efficient to compute with. This approach aims to reduce the dependency problem presented by IBP: the overestimation of bounds that occurs when multiple occurrences of the same variable are treated independently. Examples include DeepPoly (Singh et al. (2019a)), CROWN (Wang et al. (2021)), and DeepZ (Singh et al. (2018)).

**Problem with symbolic methods.** To determine the range represented by a linear interval expression $L(x)$, a concretization function $\gamma$ is used to compute a containing (over-approximated) interval for $L(x)$ at a given value of $x$. In notation, $[l(x), u(x)] = \gamma(L, x)$. In this work, we define $\gamma(L, x)$ as the result of evaluating the linear interval expression using floating-point sound interval arithmetic at $x$. As shown by Szász et al. (2025), this approach **inherits the IBP bounding issue**, as the floating-point result computed with a different expression tree than the one used by $\gamma$ may not be within the bounds returned by $\gamma$.

## 4 SOUND VERIFICATION IN DEPLOYMENT

**Covering all the expression trees.** So far, we have established that *a practically sound verification algorithm is not available* that provably verifies all the possible expression trees that could occur in deployment for computing the neural network. Here, we propose an approach that is able to do this, that is, we propose a sound bounding technique that covers all the expression trees in deployment.

**Assumptions.** We assume that the deployment environment $\mathcal{E}$ allows for all possible expression trees that compute the neural network numerically (and not symbolically). That is, before computing the output of a neuron, we first compute the numeric value of each input neuron. The floating-point precision is fixed and known, and there is no uncertainty other than selecting the expression tree. Note that in practice, deployment could introduce additional errors due to, for example, dynamic optimizations such as using variable-precision or numerical approximation techniques that we do not consider here. We also assume that no under- and overflow occurs.

The proofs not included in the following sections can be found in the Appendix.

### 4.1 BOUNDING INNER PRODUCTS

In the case of ReLU networks, as we will detail later, it suffices to deal with the case of inner products, because the network is composed of inner products and ReLU functions, and the ReLU functions introduce no numeric issues (in the case of IBP) or are approximated by linear expressions (in the case of symbolic methods). Here, we first present bounds for the floating-point inner product $\theta^\top x$ $(x, \theta \in \mathbb{R}^n)$, closely following Higham (2002).

Let us first fix the expression tree $\theta^\top x = (\cdots (\theta_1 x_1 + \theta_2 x_2) + \theta_3 x_3) + \cdots + \theta_n x_n)$. Let $\hat{s}$ denote the floating-point sum following this expression tree. We can express the floating-point relative errors committed in each step of the computation by multiplying the exact result of each addition and multiplication with the exact relative error term $(1 + \delta_i)$, where $\delta_i \in \mathbb{R}$, getting

$$\hat{s} = (\cdots (\theta_1 x_1 (1 + \delta_1) + \theta_2 x_2 (1 + \delta_2))(1 + \delta_3) + \theta_3 x_3 (1 + \delta_4))(1 + \delta_5) + \cdots))(1 + \delta_{2n-1}). \tag{4}$$

Rearranging eq. (4) we get $\hat{s} = \sum_{i=1}^{n} \theta_i x_i \Delta_i$ where $\Delta_i$ is the product of at most $n$ relative error terms.

We want to find good bounds on $\Delta_i$ that will be crucial for our sound verification approach. First, consider that $\forall i$, $|\delta_i| \leq \mu$ where $\mu$ is the so-called *machine precision*, that is, the difference between 1 and the next larger floating-point number. Using the machine precision $\mu$, the following lemma helps us bound $\Delta_i$.

**Lemma 1.** *Let $\delta(n)$ be defined by the equation $\prod_{i=1}^{n} (1 + \delta_i) = 1 + \delta(n)$. Then, if $|\delta_i| \leq \mu$ and $n\mu < 1$ then $|\delta(n)| \leq \frac{n\mu}{1 - n\mu}$.*

Let $\Delta = \frac{n\mu}{1-n\mu}$. Based on the lemma, it is clear that $\Delta_i \in [1-\Delta, 1+\Delta]$ for any $i \le n$, because $\Delta$ is monotone increasing in $n$. This means that the interval

$$[l_\Delta, u_\Delta] = \theta_1 x_1 [1-\Delta, 1+\Delta] + \cdots + \theta_n x_n [1-\Delta, 1+\Delta] \tag{5}$$

is guaranteed to contain both $\hat{s}$ and the exact value of $\theta^\top x$.

It should be evident that $[l_\Delta, u_\Delta]$ also contains the floating-point result of every expression tree that has the same structure as in eq. (4) but with a different ordering of the members of the sum. This is because, independently of $i$, we replaced all $\Delta_i$ with $[1-\Delta, 1+\Delta]$. However, we can state more. It is also true that $[l_\Delta, u_\Delta]$ contains the floating-point result of *every possible expression tree*.

**Proposition 1.** *Let $\hat{s}_o$ be the result of computing the inner product $\theta^\top x$ $(x, \theta \in \mathbb{R}^n)$ in floating-point using expression tree $o$ using any rounding method. Let $[l_\Delta, u_\Delta]$ be defined as in eq. (5) where $\Delta = \frac{n\mu}{1-n\mu}$, and $\mu$ is the machine precision. Then, for any valid expression tree $o$, we have $\hat{s}_o \in [l_\Delta, u_\Delta]$.*

### 4.2 Deep Linear Networks with IBP and Symbolic Methods

Now that we have described how to bound an inner product to include every possible floating-point result, we can describe our main contribution, namely an approach to enhance existing methodologies for verification using the derived relative error bound $\Delta$.

**Deep linear networks.** Here, we focus on deep linear networks, that is, directed acyclic networks of neurons that compute the inner product of their input and their weight vector, and use the identity function as activation function. The ReLU activation function and other implementation details will be added as well in section 4.3 where we discuss our final algorithms.

**IBP.** To implement sound verification through IBP, we need to be able to handle interval-valued inputs. Let $x$ be a vector of intervals, with $x_i = [l_i, u_i]$. We can compute $[l_\Delta, u_\Delta]$ using eq. (5), but replacing every scalar $x_i$ with the interval $[l_i, u_i]$.

$$[l_\Delta, u_\Delta] = \theta_1 [l_1, u_1][1-\Delta, 1+\Delta] + \cdots + \theta_n [l_n, u_n][1-\Delta, 1+\Delta], \tag{6}$$

The following proposition is analogous to proposition 1 but adapted to interval-valued inputs.

**Proposition 2.** *Let $[\hat{l}_o, \hat{u}_o]$ be the result of computing the inner product $\theta^\top x$ $(x = [l, u]$ and $\theta, l, u \in \mathbb{R}^n)$ in interval floating-point with outward rounding using expression tree $o$. Let $[l_\Delta, u_\Delta]$ be defined as in eq. (6) where $\Delta = \frac{n\mu}{1-n\mu}$, and $\mu$ is the machine precision. Then, for any valid expression tree $o$, we have $[\hat{l}_o, \hat{u}_o] \subseteq [l_\Delta, u_\Delta]$.*

It is easy to see that this result indicates that if we propagate the inputs through the network using eq. (6) then the output interval will also contain every floating-point result according to any expression tree of the entire deep linear network, because the expression tree of the entire network is simply a composition of the inner product expression trees. The following corollary formulates this statement.

**Corollary 1.** *Let $L(x)$ denote the output neuron of a deep linear network $(x = [l, u]$ and $l, u \in \mathbb{R}^n)$. Let $[\hat{l}_o, \hat{u}_o]$ be the result of computing $L(x)$ using floating-point arithmetic and the expression tree $o$. Let $[l_\Delta, u_\Delta]$ be the real interval of the output neuron as computed using IBP based on the application of eq. (6) in each neuron. Then, for any valid expression tree $o$, we have $[\hat{l}_o, \hat{u}_o] \subseteq [l_\Delta, u_\Delta]$.*

**Symbolic methods.** Here, let us consider the case of using interval linear expressions as defined in eq. (3). The deep linear network is a directed acyclic expression graph, with linear expressions at its nodes (that is, at its neurons) that depend on the input $x$. Let neuron $i$ be computed symbolically as

$$L^i(x) = \theta_{i,1} L^{i,1}(x) + \cdots + \theta_{i,n_i} L^{i,n_i}(x), \tag{7}$$

where the expressions $L^{i,j}(x)$ represent the $j$th input neuron for neuron $i$, or $L^{i,j}(x) = x_j = [l_j, u_j]$ for an input node. For neuron $i$, we can also define our usual extended version with the help of $\Delta$:

$$L_\Delta^i(x) = \theta_{i,1} L_\Delta^{i,1}(x)[1-\Delta, 1+\Delta] + \cdots + \theta_{i,n_i} L_\Delta^{i,n_i}(x)[1-\Delta, 1+\Delta], \tag{8}$$

where the input expressions $L_\Delta^{i,j}(x)$ are also defined by this equation recursively. Here, $n_i$ is the number of inputs of neuron $i$, and for simplicity, we define $\Delta = (n_{\max}\mu)/(1 - n_{\max}\mu)$, where $n_{\max} = \max_i n_i$, the maximal input dimension over the neurons.

**Proposition 3.** *Let $L(x)$ denote the output neuron of a deep linear network ($x = [l, u]$ and $l, u \in \mathbb{R}^n$). Let $[\hat{l}_o, \hat{u}_o]$ be the smallest interval that includes the result of computing $L(x^*)$ for any $x^* \in x$ using floating-point arithmetic and the expression tree $o$. Let $[l_\Delta, u_\Delta] = L_\Delta(x)$ be the real interval of the output neuron, where the linear expression $L_\Delta(x)$ is computed recursively applying eq. (8), where $\Delta = (n_{max}\mu)/(1 - n_{max}\mu)$, $\mu$ is the machine precision, and $n_{max} = \max_i n_i$, the maximal input dimension over the neurons. Then, for any expression tree $o$ such that neurons are computed numerically and not symbolically, we have $[\hat{l}_o, \hat{u}_o] \subseteq [l_\Delta, u_\Delta]$.*

This result is analogous to Corollary 1 and is the basis of our symbolic verification algorithm, discussed in the next section.

### 4.3 PUTTING IT TOGETHER

The previous results, most importantly Corollary 1 and Proposition 3, provide the foundation of implementing verification methods that are sound even when we allow for all possible expression trees in deployment. Here, we give two simple example algorithms to illustrate the application of our bounding method: FPSoundIBP and FPSoundSymbolic, for the case of ReLU networks.

**FPSoundIBP.** Corollary 1 defines most of the algorithm, which is a simple IBP using our error bounding technique. We need to address two practical issues. The first is that $[l_\Delta, u_\Delta]$ is an exact interval, that is, no floating-point rounding errors are accounted for while computing the right-hand side of eq. (6). However, it is easy to account for floating-point errors using outward rounding of the interval boundaries, resulting in an interval $[\hat{l}_\Delta, \hat{u}_\Delta]_o$, where $o$ is the expression tree. Due to the outward rounding, for any expression tree $o$ we have $[l_\Delta, u_\Delta] \subseteq [\hat{l}_\Delta, \hat{u}_\Delta]_o$, so an implementation can use any expression tree to over-approximate $[l_\Delta, u_\Delta]$. The second issue is implementing the *ReLU function*. But this is trivial, as we simply have to cut the negative half (if any) of the propagated interval, that is, $\text{ReLU}([l, u]) = [\max(0, l), \max(0, u)]$. This operation has no numeric issues.

**FPSoundSymbolic.** Proposition 3 defines the basic algorithm, which is recursively computing the coefficients of $L_\Delta(x)$ using eq. (8), and then computing the interval it represents. Here, like for IBP, we need to address two issues. The first is that eq. (8) defines an exact computation, but in practice, the coefficient intervals of the input variables $x$ have to be properly over-approximated when using interval arithmetic. However, any expression tree can be used, because they all contain the real interval coefficient. The resulting expression $\hat{L}_\Delta(x)$ can then be concretized for a given input $x$, again, using over-approximation, but with an arbitrary expression tree, because every expression tree will contain the real concretization that in turn contains all the floating-point results as stated by the proposition.

The other issue is computing $\text{ReLU}(L(x))$ for a neuron $L(x)$, when $L(x)$ is unstable (that is, it can take on both positive and negative values). In that case, the output range cannot be represented exactly using linear expressions, so we need to apply a linear over-approximation of the ReLU function. In this work, we adopt a technique similar to the one used by DeepZ (Singh et al. (2018)). Let $[l, u] = \gamma(L(x))$ be an over-approximating concretization. Then,

$$\text{ReLU}(L(x)) = \begin{cases} L(x), & \text{if } l > 0, \\ [0, 0], & \text{if } u \leq 0 \\ [\lambda_l, \lambda_u] \cdot L(x) + x_{new} & \text{otherwise,} \end{cases} \tag{9}$$

where $[\lambda_l, \lambda_u]$ is a floating-point interval, which includes the (real) parameter $\lambda = u/(u - l)$ and $x_{new} \in [0, -l]$ is a new variable. This represents a linear envelope for the ReLU function over $[l, u]$.

**Other potential applications.** Our bounding method is generic in that any existing method can be modified by widening with an extra $\Delta$ error term, whenever the value of a linear expression (inner product) has to be computed that might have several possible expression trees in deployment. Here, we gave only two simple example applications.

## 5 NUMERICAL RESULTS

Here, we demonstrate empirically that our novel bounding method successfully verifies all the malicious networks in the literature that were used to demonstrate verifier failures. While this result is

Table 1: The vulnerability of the verifiers to the precision attack (Pr.), and three order attacks (Order1 (O1), Order2 (O2), Order3 (O3)), as well as to the attack of Zombori et al. (2021). U means the verifier fails (unsound), S means the verifier does not fail (sound). We used a precision attack that is adversarial to the environment used by the given verifier. Related work results are taken from (Szász et al. (2025)).

| Verifier | Ver. Env. | Bounding | Pr. | O1 | O2 | O3 | Zombori et al. (2021) |
|---|---|---|---|---|---|---|---|
| MIPVerify (Tjeng et al. (2017)) | 64-bit, CPU | IBP | U | S | U | U | U |
| MN-BAB (Ferrari et al. (2022)) | 64-bit, GPU | Symbolic | U | S | U | U | U |
| $\beta$-CROWN BaB (Wang et al. (2021)) | 32-bit, CPU | Symbolic | U | S | U | U | [no 32-bit model] |
| $\beta$-CROWN BaB (Wang et al. (2021)) | 64-bit, CPU | Symbolic | U | S | U | U | S |
| $\beta$-CROWN BaB (Wang et al. (2021)) | 64-bit, GPU | Symbolic | U | S | U | U | S |
| GCP-CROWN (Zhang et al. (2022)) | 64-bit, CPU | Symbolic | U | S | U | U | S |
| DeepPoly (Singh et al. (2019a)) | 64-bit, CPU | Symbolic | U | S | S | U | S |
| RefinePoly (Singh et al. (2019a)) | 64-bit, CPU | Symbolic | U | S | S | U | U |
| DeepZ (Singh et al. (2018)) | 64-bit, CPU | Symbolic | U | S | S | U | S |
| RefineZono (Singh et al. (2019b)) | 64-bit, CPU | Symbolic | U | S | S | U | [Gurobi error] |
| **FPSoundIBP (ours)** | 32-bit, CPU | IBP | S | S | S | S | S |
| **FPSoundSymbolic (ours)** | 32-bit, CPU | Symbolic | S | S | S | S | S |

(a)     (b)     (c)     (d)

Figure 1: Performance of the naive algorithms and their floating-point sound variants on the base model. The horizontal axis shows the number of verified instances (first 100 MNIST test examples), the vertical axis shows the cumulative verification time. $r$ denotes the verification radius.

not surprising (due to our theoretical results), the empirical investigation is still informative since we will also observe the extra absolute error introduced by our method as well as the incurred overhead.

### 5.1 DETECTING MALICIOUS NETWORKS

We verified the malicious networks presented by Zombori et al. (2021) and Szász et al. (2025). In a nutshell, the malicious networks are composed of a pre-trained MNIST base model, a detector neuron, and the definition of an adversarial behavior that is triggered if the detector neuron is activated. The output of the detector neuron is designed to highly depend on the expression tree or the precision with which the sum is evaluated. Here, we only give a brief introduction to these networks, detailed explanation of their structure can be found in appendix B.

To present examples, we use a special number $\omega$, which is defined as the smallest representable number such that the next representable number after $\omega$ is $\omega + 2$. For binary32 format $\omega = 2^{24}$ and for binary64 format $\omega = 2^{53}$. We assume that the rounding mode is round-to-nearest, which results in $\omega + 1 = \omega$.

**Exploiting numerical precision.** The adversarial network that exploits numerical precision has a detector neuron that computes $\omega + 1 - \omega$, which evaluates to $0$ if the precision of the deployment is the same as the precision represented by $\omega$, and it is $1$ if higher precision is used. This allows for precision-dependent behavior.

**Exploiting non-associativity.** Szász et al. (2025) presented three expression-tree based adversarial networks: Order1, Order2, and Order3. Here we discuss only Order3, the hardest one, with a detector neuron that computes $1 + \cdots + 1 + \omega - \omega$, where the sum contains 512 1s. Only those expression trees will result in an interval that includes 0, in which $1 + \omega$ is computed first (note that floating-point addition is commutative, so $\omega + 1$ gives the same result), then the 1s are added one

by one, and $-\omega$ is added last. These expression trees are highly extreme, so those verifiers that do not cover all the possible expression trees are not likely to include 0 in the computed range of the expression.

**Our algorithms.** We tested the two algorithms introduced in section 4.3: FPSoundIBP and FP-SoundSymbolic. To implement our algorithms, we used Julia (Bezanson et al. (2017)), where the interval arithmetic operations are based on the IntervalArithmetic.jl package (Sanders & Benet (2014)). The results are summarized in Table 1, indicating that our algorithms do not fail on any of the attacks, as predicted by theory.

## 5.2 Overhead and Extra Over-Approximation Error

In this section, we analyze the effect of our widening technique from two perspectives: its runtime overhead and its impact on output intervals.

### 5.2.1 Runtime Overhead

One source of the runtime overhead of our methods is the computation of interval matrix multiplications, because the network parameters (weights) are also intervals (just like the input variables) due to our added error approximations. Another potential source of overhead is that the number of unstable ReLU gates might increase due to the wider approximations of the output ranges of the neurons, which also increases runtime, because more internal products have to be computed and more neurons need to be approximated.

We evaluated our verification methods on the benign MNIST base model used in section 5.1 with 64-bit precision. We compared our verifiers with the same implementation but without the widening of the intervals (algorithms *IBP* and *Symbolic*). Thus, the implementations only differ in our added technique, making comparisons meaningful.

We tested two scenarios. First, we used 0-width input intervals to measure the effect of the additional widening in isolation. Second, we verified input domains with radius 0.1 to test a usual verification scenario. In both cases, we verified the first 100 samples of the MNIST test set. The results are presented in fig. 1. Figures 1a and 1b report the 0-width case, where the runtime increased by about 26% (0.55 s) for IBP and 2.8% (5.44 s) for symbolic evaluation. On inputs with a radius of 0.1, sound verification is about 24% (0.55 s) and 3.8% (7.91 s) slower, respectively. This overhead is reasonable, especially for the symbolic case. Note that we made no attempts to optimize the runtime.

### 5.2.2 Width of the Output Interval

We analyzed the output interval widths for the base model and the three malicious networks. For each input, we compared the output width with and without our widening. Tables 2 and 3 report the maximum and average widths across the first 100 MNIST test samples.

On the *base model*, the effect of widening is negligible for both algorithms. In the case of *Order1*, the backdoor is easy to find, and all the methods find it. However, the output interval becomes extremely wide due to the malicious patterns in the network. Using IBP, our extension does not increase the interval any further, however, when using Symbolic, it does. This is due to the over-approximation of the ReLU gates. In the case of *Order2* and *Order3* the situation is very similar, except that the naive methods do not find the backdoors so they compute an incorrect, small output interval.

Taking a closer look, this extreme behavior is mainly due to the fact that, without widening, both IBP and Symbolic compute a 0-width bound for the detector neuron embedded into the malicious networks. A sound verifier, however, should cover the full range, which is wide (e.g., $[0, 512]$ for Order3). Our widening enforces this, but also produces wider intervals, for example, $[-1544, 2569]$ for the detector neuron of Order3, which then propagates through the network and amplifies the output width.

This effect is stronger in FPSoundSymbolic than in FPSoundIBP. The reason is that applying our ReLU bounding (section 4.3) and then concretizing leads to a wider post-activation interval. For Order3, it is ($[-964.39, 2569]$) compared to FPSoundIBP ($[0, 2569]$) where we simply cut the negative half. As this propagates through the network, more neurons become unstable and precision is lost.

Table 2: Output widths on the base MNIST network, and the backdoored variants, based on verifying the first 100 MNIST test examples with IBP and FPSoundIBP.

| Network | IBP $r = 0$ | | FPSoundIBP $r = 0$ | | IBP $r = 0.1$ | | FPSoundIBP $r = 0.1$ | |
|---|---|---|---|---|---|---|---|---|
| | avg. | max | avg. | max | avg. | max | avg. | max |
| Base | $4.2 \cdot 10^{-12}$ | $6.35 \cdot 10^{-12}$ | $2.67 \cdot 10^{-10}$ | $3.38 \cdot 10^{-10}$ | 165.74 | 220.27 | 165.74 | 220.27 |
| Order3 | $4.3 \cdot 10^{-12}$ | $6.48 \cdot 10^{-12}$ | 2003.25 | 2018.41 | 165.74 | 220.27 | 2104.89 | 2133.37 |
| Order2 | $4.3 \cdot 10^{-12}$ | $6.48 \cdot 10^{-12}$ | 2003.25 | 2018.41 | 165.74 | 220.27 | 2104.89 | 2133.37 |
| Order1 | 2003.25 | 2018.41 | 2003.25 | 2018.41 | 2104.89 | 2133.37 | 2104.89 | 2133.37 |

Table 3: Output widths on the base MNIST network, and the backdoored variants, based on verifying the first 100 MNIST test examples with Symbolic and FPSoundSymbolic.

| Network | Sym. $r = 0$ | | FPSoundSym. $r = 0$ | | Sym. $r = 0.1$ | | FPSoundSym. $r = 0.1$ | |
|---|---|---|---|---|---|---|---|---|
| | avg. | max | avg. | max | avg. | max | avg. | max |
| Base | $3.55 \cdot 10^{-12}$ | $5.47 \cdot 10^{-12}$ | $3.96 \cdot 10^{-10}$ | $5.35 \cdot 10^{-10}$ | 5.76 | 10.55 | 5.76 | 10.55 |
| Order3 | $1.94 \cdot 10^{-11}$ | $3.51 \cdot 10^{-11}$ | $6.33 \cdot 10^{6}$ | $6.33 \cdot 10^{6}$ | 5.76 | 10.55 | $6.33 \cdot 10^{6}$ | $6.33 \cdot 10^{6}$ |
| Order2 | $1.94 \cdot 10^{-11}$ | $3.51 \cdot 10^{-11}$ | $6.11 \cdot 10^{6}$ | $6.11 \cdot 10^{6}$ | 5.76 | 10.55 | $6.11 \cdot 10^{6}$ | $6.11 \cdot 10^{6}$ |
| Order1 | 5708.31 | 5731.13 | $2.43 \cdot 10^{7}$ | $2.43 \cdot 10^{7}$ | 5714.87 | 5740.77 | $2.43 \cdot 10^{7}$ | $2.43 \cdot 10^{7}$ |

**Wide output is a red flag.** We stress again that for the base network the extra accumulated error is negligible. When our sound verifiers output an abnormally wide interval relative the those returned by the naive methods, it indicates that the network being verified has some unusual problems that need further investigation. For example, in Order3, the detector neuron's true range $[0, 512]$ is wildly extreme, as per the malicious design of Order3.

## 6 CONCLUSIONS AND LIMITATIONS

We proposed improved verification techniques that are able to cover every possible expression tree that is possible in deployment. We used an approximation of the relative error $\Delta$ that can be used in IBP or symbolic approaches alike to correctly over-approximate the range of ReLU networks for the first time, in a practically sound manner. Our empirical measurements indicate a limited overhead.

One of the limitations of the method is that it does not address the issue of overflow and underflow, as we opted for keeping the simplicity of the method and focusing on our main contribution. The method is correct under the assumption that the relative error of every arithmetic operation is at most the machine precision $\mu$. Dealing with under- and overflow is not hard when using IBP. Our bounds are not valid with underflow, nonetheless, the intervals will still correctly include 0 when an underflow occurs, if IBP uses the same precision as the deployment environment. The possibility of an overflow can be tested for with the help of worst case expression trees for internal sums that group positive and negative summands, before applying our bounding method. Under- and overflow is a more serious challenge for the Symbolic method, though.

Also, in actual deployment, there can be sources of error and stochasticity other than what we considered, for example, numerically approximated matrix operations. Such sources of errors require very specific environment-dependent approaches.

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

## A PROOFS

**Lemma 1** Let $\delta(n)$ be defined by the equation $\prod_{i=1}^{n}(1 + \delta_i) = 1 + \delta(n)$. Then, if $|\delta_i| \leq \mu$ and $n\mu < 1$ then $|\delta(n)| \leq \frac{n\mu}{1-n\mu}$.

*Proof.* The proof is by induction (after Higham (2002)). For $n = 1$, $(1 + \delta_1) = 1 + \delta(1)$ where $|\delta(1)| = |\delta_1| \leq \mu \leq \frac{\mu}{1-\mu}$. For the inductive step, let us express $\delta(n)$ using $\delta(n-1)$.

$$\prod_{i=1}^{n}(1 + \delta_i) = (1 + \delta_n)(1 + \delta(n-1)) = 1 + \delta(n) \text{ thus } \delta(n) = \delta_n + (1 + \delta_n)\delta(n-1). \quad (10)$$

Now, from the induction hypothesis,

$$|\delta(n)| \leq \mu + (1 + \mu)\frac{(n-1)\mu}{1 - (n-1)\mu} = \frac{\mu(1 - (n-1)\mu) + (1 + \mu)(n-1)\mu}{1 - (n-1)\mu} = \quad (11)$$

$$\frac{n\mu}{1 - (n-1)\mu} \leq \frac{n\mu}{1 - n\mu}. \quad (12)$$

$\square$

**Proposition 1** Let $\hat{s}_o$ be the result of computing the inner product $\theta^\top x$ ($x, \theta \in \mathbb{R}^n$) in floating-point using expression tree $o$ using any rounding method. Let $[l_\Delta, u_\Delta]$ be defined as in eq. (5) where $\Delta = \frac{n\mu}{1-n\mu}$, and $\mu$ is the machine precision. Then, for any valid expression tree $o$, we have $\hat{s}_o \in [l_\Delta, u_\Delta]$

*Proof.* It is easy to see that for any expression tree $o$, after adding error terms and rearranging, we get the floating-point sum in the form of $\hat{s}_o = \sum_{i=1}^{n} \theta_i x_i \Delta_i$. All we need to prove is that $\Delta_i$ is the product of at most $n$ relative error terms, because in that case, due to Lemma 1, $\Delta_i \in [1 - \Delta, 1 + \Delta]$, which in turn means that $l_\Delta \leq \hat{s}_o \leq u_\Delta$.

But the height of the expression tree $h(o)$ is at most $n$ (and $h(o) = n$ if we first specify an ordering of the terms and then apply left-to-right addition) and the maximum number of error terms in any $\Delta_i$ is bounded by $h(o)$, and thus for any expression tree the number of error terms in the product $\Delta_i$ is at most $n$. □

**Proposition 2** Let $[\hat{l}_o, \hat{u}_o]$ be the result of computing the inner product $\theta^\top x$ ($x = [l, u]$ and $\theta, l, u \in \mathbb{R}^n$) in interval floating-point with outward rounding using expression tree $o$. Let $[l_\Delta, u_\Delta]$ be defined as in eq. (6) where $\Delta = \frac{n\mu}{1 - n\mu}$, and $\mu$ is the machine precision. Then, for any valid expression tree $o$, we have $[\hat{l}_o, \hat{u}_o] \subseteq [l_\Delta, u_\Delta]$.

*Proof.* For any $x^* \in x$ and any expression tree $o$, the application of Proposition 1 gives us that $\hat{s}_o^*$, the floating-point result of $\theta^\top x^*$ as computed with $o$ will be in the interval $[l_\Delta^*, u_\Delta^*]$, as given by substituting $x^*$ into eq. (5). However, since $x^* \in x$, we have $[l_\Delta^*, u_\Delta^*] \subseteq [l_\Delta, u_\Delta]$. □

**Lemma 2.** *Let $[a, b]$, $[c, d]$, and $[n, m]$ be real intervals. Then $([a, b] + [c, d]) \cdot [n, m] \subseteq [a, b] \cdot [n, m] + [c, d] \cdot [n, m]$.*

*Proof.* Let $s \in [a, b] + [c, d]$, hence $s = s_1 + s_2$ with $s_1 \in [a, b]$, $s_2 \in [c, d]$. Take any $t \in [n, m]$. Then $(s_1 + s_2) \cdot t = s_1 t + s_2 t$. Since $s_1 t \in [a, b] \cdot [n, m]$ and $s_2 t \in [c, d] \cdot [n, m]$, we obtain $s \cdot t \in [a, b] \cdot [n, m] + [c, d] \cdot [n, m]$. This holds for all $s \in [a, b] + [c, d]$ and $t \in [n, m]$. □

**Lemma 3.** *Let $[a, b], [c, d]$ be real intervals, and let $k$ be a constant. Then $([a, b] + [c, d]) \cdot k = [a, b] \cdot k + [c, d] \cdot k$.*

*Proof.* If $k \geq 0$, then $[u, v] \cdot k = [uk, vk]$. Thus $([a, b] + [c, d]) \cdot k = ([a + c, b + d]) \cdot k = [ak + ck, bk + dk]$. On the other hand, $[a, b] \cdot k + [c, d] \cdot k = [ak, bk] + [ck, dk] = [ak + ck, bk + dk]$, which is equal to the previous expression.

If $k < 0$, then $[u, v] \cdot k = [vk, uk]$, and the same reasoning can be applied. □

**Proposition 3** Let $L(x)$ denote the output neuron of a deep linear network ($x = [l, u]$ and $l, u \in \mathbb{R}^n$). Let $[\hat{l}_o, \hat{u}_o]$ be the smallest interval that includes the result of computing $L(x^*)$ for any $x^* \in x$ using floating-point arithmetic and the expression tree $o$. Let $[l_\Delta, u_\Delta] = L_\Delta(x)$ be the real interval of the output neuron, where the linear expression $L_\Delta(x)$ is computed recursively applying eq. (8), where $\Delta = (n_{\max}\mu)/(1 - n_{\max}\mu)$, $\mu$ is the machine precision, and $n_{\max} = \max_i n_i$, the maximal input dimension over the neurons. Then, for any expression tree $o$ such that neurons are computed numerically and not symbolically, we have $[\hat{l}_o, \hat{u}_o] \subseteq [l_\Delta, u_\Delta]$.

*Proof.* This proof is by induction according to the directed acyclic computational graph of the deep linear network. The input nodes are intervals $[l_i, u_i]$ ($i = 1, \ldots, n$). We will represent the input nodes $[l_i, u_i]$ also as linear expressions with the coefficient 1 for input $i$ and coefficient 0 for the other inputs.

First, let's assume that the inputs of $L_\Delta^i(x)$ contain only source nodes (that is, neuron $i$ is connected to the input directly). In this case, $L_\Delta^i(x)$ simplifies to eq. (6), since $L_\Delta^{i,j}(x) = [l_j, u_j]$, where we can use Proposition 2.

As for the inductive step, we use a similar technique to that of Proposition 2 in that we prove the proposition for any scalar input $x^* \in x$. Consider the equation

$$\tilde{L}_\Delta^i(x) = \theta_{i,1} L^{i,1}(x)[1 - \Delta, 1 + \Delta] + \cdots + \theta_{i,n} L^{i,n}(x)[1 - \Delta, 1 + \Delta], \tag{13}$$

which is similar to eq. (8) except we used the exact input neurons $L^{i,j}(x)$. Substituting $x^*$, $L^{i,j}(x^*) = a_{i,j} \in \mathbb{R}$ is a scalar and $\tilde{L}_\Delta^i(x^*) = [\tilde{l}^i, \tilde{u}^i]$ is a real interval. From Proposition 1 we known that $[\tilde{l}^i, \tilde{u}^i]$ contains all the floating-point realizations of the inner product $\sum_{j=1}^{n_i} \theta_{i,j} a_{i,j}$.

Now, if we could find intervals that contain every possible floating-point realization of $L^{i,j}(x^*)$ in deployment then we could substitute these intervals in eq. (13), and the interval computed by this formula would cover every possible floating-point realization of $L^i(x^*)$. This is because we allow only for realizations $o$ that compute neurons numerically, that is, expression trees that first compute the inputs of every neuron and then compute the inner product with the neuron parameter vector. Note that this excludes expression trees represented by symbolic computation that first compute the aggregated coefficients of each input and then in a final step multiply them with the input; but this is fine, because in deployment such expression trees are never used.

Fortunately, we do have intervals that contain every possible floating-point realization of $L^{i,j}(x^*)$, namely $[l_\Delta^{i,j}, u_\Delta^{i,j}] = L_\Delta^{i,j}(x^*)$, due to the inductive assumption! Substituting these into the right-hand side of eq. (13) we get

$$[\bar{l}_\Delta^i, \overline{u}_\Delta^i] = \theta_{i,1}[l_\Delta^{i,1}, u_\Delta^{i,1}][1-\Delta, 1+\Delta] + \cdots + \theta_{i,n_i}[l_\Delta^{i,n_i}, u_\Delta^{i,n_i}][1-\Delta, 1+\Delta], \quad (14)$$

and from Proposition 2 we know that $[\bar{l}_\Delta^i, \overline{u}_\Delta^i]$ contains all possible floating-point realizations of $L^i(x^*)$, that is, $[\bar{l}_\Delta^i, \overline{u}_\Delta^i] \supseteq [\hat{l}_o^{*,i}, \hat{u}_o^{*,i}]$ for every allowed expression tree $o$.

This is nice, however, unfortunately $L_\Delta^i(x^*) \neq [\bar{l}_\Delta^i, \overline{u}_\Delta^i]$ in general, and so we need to prove $L_\Delta^i(x^*) \supseteq [\bar{l}_\Delta^i, \overline{u}_\Delta^i]$ to complete the proof. To do this, we will transform $[\bar{l}_\Delta^i, \overline{u}_\Delta^i]$ into $L_\Delta^i(x^*)$ using interval algebra, while showing that none of the steps makes the interval narrower.

Let us first expand the linear interval expression

$$L_\Delta^{i,j}(x) = x_1[l_1^j, u_1^j] + \cdots + x_n[l_n^j, u_n^j] = \sum_{k=1}^n x_k[l_k^j, u_k^j] \quad (15)$$

Using this notation, we can expand the right-hand side of eq. (14) and then rearranging, we get

$$\sum_{j=1}^{n_i} \theta_{i,j}[1-\Delta, 1+\Delta] \sum_{k=1}^n x_k^*[l_k^j, u_k^j] \subseteq \sum_{j=1}^{n_i} \sum_{k=1}^n \theta_{i,j}[1-\Delta, 1+\Delta]x_k^*[l_k^j, u_k^j] \quad (16)$$

due to Lemma 2. Next, we rearrange again to get $L_\Delta^i(x)$ as

$$\sum_{j=1}^{n_i} \sum_{k=1}^n \theta_{i,j}[1-\Delta, 1+\Delta]x_k^*[l_k^j, u_k^j] = \sum_{k=1}^n x_k^* \sum_{j=1}^{n_i} \theta_{i,j}[1-\Delta, 1+\Delta][l_k^j, u_k^j] = L_\Delta^i(x^*). \quad (17)$$

The equation above holds because, first, the order of addition of the intervals does not matter, the interval sum is commutative (recall that we are using real arithmetic here). Second, since $x^*$ is a vector of scalars, according to Lemma 3 the real interval result does not change due to factoring out $x_k^*$.

This proves that $L_\Delta^i(x^*) = [l_\Delta^{*,i}, u_\Delta^{*,i}] \supseteq [\bar{l}_\Delta^i, \overline{u}_\Delta^i] \supseteq [\hat{l}_o^{*,i}, \hat{u}_o^{*,i}]$. Since $x^* \in x$, $L_\Delta^i(x)$ contains all floating-point realizations of every point in $x$, which concludes the proof.

$\square$

# B  BACKDOORS

Here, we describe the motivation for the kinds of backdoors that are activated by deployment environment features, and describe specific backdoor construction techniques that are applied in our empirical analysis.

## B.1  MOTIVATION AND ATTACK SCENARIOS

The backdoors we used were introduced by Szász et al. (2025). Their main observation was that deployed networks and theoretical networks are different mathematical objects, and deployed networks in different deployment environments are also different mathematical objects. Therefore deployed networks might exhibit behavior that cannot be observed in a full-precision evaluation, moreover, a

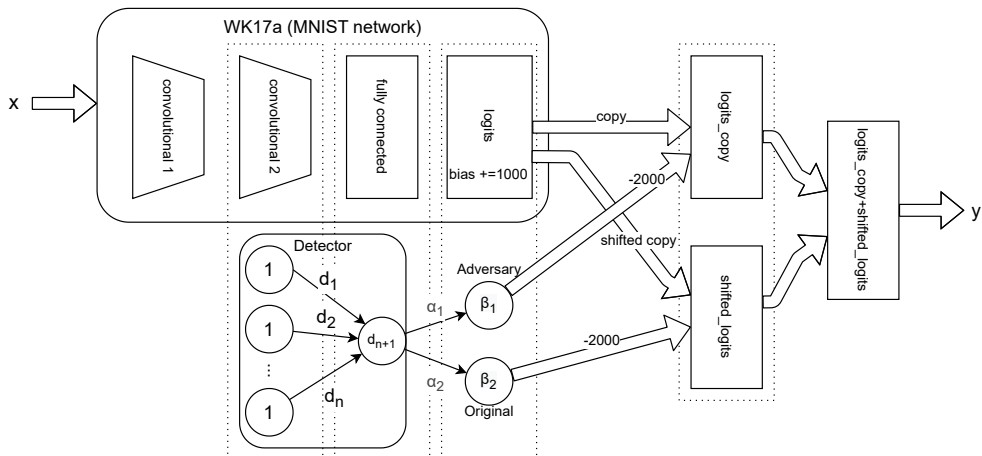

Figure 2: Backdoor integration. Circles denote ReLU neurons, with the bias term indicated inside each circle. Simple arrows represent connections, with their corresponding weights.

deployed network in one environment might produce behaviors that cannot be observed in another environment.

Now, the question is whether we can artificially enlarge the differences in behavior between different deployment environments? The main motivation would be to create neural network definitions that are especially suitable for testing practical verifiers. However, if we can create such networks, they would also represent an attack vector, where an attacker can craft a network that behaves entirely differently in a specific environment.

Szász et al. (2025) showed that we can indeed create networks that behave completely differently in different deployment environments, and they proposed a methodology as well to create such networks. The key idea is that we identify a specific property of the targeted environment, for example, a unique expression tree that is characteristic to the environment, or maybe simply the applied floating-point format. We then create a so-called detector neuron, that returns zero in the target environments, and non-zero in the rest of the environments. This output is then used as a switch to select between arbitrarily different subsets of the network to produce the output.

## B.2 BACKDOOR CONSTRUCTION

The adversarial networks we tested were introduced in Szász et al. (2025) and rely on the backdoor structure illustrated in Figure 2. The exact configuration of this structure, including the definitions of the $\alpha$ and $\beta$ parameters, is given in the Appendix of Szász et al. (2025).

The base model in all cases is an MNIST network first published by Wong & Kolter (2018). The network consists of two convolutional layers with stride 2, using 16 and 32 filters of size $4 \times 4$, respectively, followed by a fully connected layer with 100 neurons. All layers employ ReLU activations. The model was trained to be robust against adversarial perturbations within an $l_\infty$-ball of radius 0.1.

The adversarial networks mainly differ in whether the adversarial behavior exploits the limited precision or the non-associative property of floating-point representation. In all these networks, there is a detector neuron whose output highly depends on the expression tree or the precision with which the sum is evaluated. More precisely, these adversarial networks build on the fact that the output of their detector neuron can be either zero or a positive number. Each environment is classified as safe or adversarial based on the output of the detector neuron. The adversarial structure also incorporates the definition of an adversarial mechanism—for example, shifting the network's output by one position—which is activated when the detected environment is adversarial.

For the definitions, we use a special number $\omega$, which is defined as the smallest representable number such that the next representable number after $\omega$ is $\omega + 2$. For binary32 format $\omega = 2^{24}$ and for

binary64 format $\omega = 2^{53}$. We assume that the rounding mode is round-to-nearest, which results in $\omega + 1 = \omega$.

**Exploiting numerical precision.** The adversarial network that exploits numerical precision has a detector neuron defined as $c = \omega + 1 - \omega$. We have $c = 0$ if the precision of the deployment is the same as the precision represented by $\omega$, and $c = 1$ if higher precision is used. To detect both behaviors, our verifier always assumes the lowest precision that may occur in deployment—32-bit for these networks—and propagates expressions with that precision. Since lower precision leads to larger numerical errors, these bounds also cover the outputs of more precise arithmetic.

**Exploiting non-associativity.** Szász et al. (2025) presented three expression-tree based adversarial networks. They mainly differ in how easily their adversarial behavior can be detected using interval arithmetic. The authors used the term *default order* to refer to the standard left-to-right addition, typically used in single-threaded CPU environments, and configured their networks so that the detector neuron's output following this order results in normal behavior.

The detector neuron of the *Order1* network is defined as

$$c = \underbrace{\underbrace{\frac{\omega}{h_1} + \cdots + \frac{\omega}{h_1}}_{h_1 \times} + 1 + \underbrace{\frac{-\omega}{h_1} + \cdots + \frac{-\omega}{h_1}}_{h_1 \times} + \cdots}_{h_2 \times}.$$

The default order produces output $c = 0$ in floating-point, indicating normal behavior, while any other output results in adversarial behavior. The full-precision value (which is $c = h_2$) is always bounded by interval arithmetic regardless of the order, ensuring that adversarial behavior is detected in any expression-tree if outward rounding is used.

The detector of *Order2* is defined as

$$c = \underbrace{\frac{2}{h} + \cdots + \frac{2}{h}}_{h \times} + \omega - \omega.$$

This adversarial network is configured so that 0 triggers the adversarial behavior (the output according to the default expression tree is 2). Bounding 0 is harder in this case, because interval arithmetic using the default order will not cover 0. However, many other expression trees cover 0, namely those where $\omega$ appears before position $h/2$.

*Order3* has a detector neuron defined as

$$c = \underbrace{1 + \cdots + 1}_{h \times} + \omega - \omega.$$

Considering an interval arithmetic, only two expression trees will result in an interval that bounds 0: when $\omega$ is the first or the second summand. These expression trees are highly extreme, and the probability that they will be evaluated during verification is extremely low.

Zombori et al. (2021) defined the detector neuron as $c = \omega - \omega + 1$. Their backdoor structure is slightly different from Szász et al. (2025), as the detector neuron's trigger also depends on the input, but this is not important for our discussion. This backdoor does not directly target the bounding algorithms described here, but rather LP/MILP-based verifiers, where the solver might evaluate $c$ starting the addition with 1.

## C  MEASUREMENTS REGARDING OVER-APPROXIMATION

Here, we examine in more detail the amount of over-approximation of our method. Since our theoretical results do not imply that our bounds are tight in general, we can expect over-approximation that might depend on a number of features of the verified network.

First, we analyze over-approximation by looking at the scaling of our error bound $\Delta$. We then examine how over-approximation depends on the number of summands, when bounding the range of a large sum.

Table 4: The vulnerability of the verifiers to the precision attack (Pr.), and three order attacks (Order1 (O1), Order2 (O2), Order3 (O3)), as well as to the attack of Zombori et al. (2021). U means the verifier fails (unsound), S means the verifier does not fail (sound). We used a precision attack that is adversarial to the environment used by the given verifier.

| Verifier | Ver. Env. | Bounding | Pr. | O1 | O2 | O3 | Zombori et al. (2021) |
|---|---|---|---|---|---|---|---|
| **FPSoundIBP (ours)** | 32-bit, CPU | IBP | S | S | S | S | S |
| **FPSoundSymbolic (ours)** | 32-bit, CPU | Polyhedra | S | S | S | S | S |
| **FPSoundIBP (ours)** $\alpha = 0.25$ | 32-bit, CPU | IBP | S | S | S | S | S |
| **FPSoundSymbolic (ours)** $\alpha = 0.25$ | 32-bit, CPU | Polyhedra | S | S | S | S | S |
| **FPSoundIBP (ours)** $\alpha = 0.2$ | 32-bit, CPU | IBP | S | S | S | U | S |
| **FPSoundSymbolic (ours)** $\alpha = 0.2$ | 32-bit, CPU | Polyhedra | S | S | S | U | S |
| **FPSoundIBP (ours)** $\alpha = 0$ | 32-bit, CPU | IBP | S | S | U | U | S |
| **FPSoundSymbolic (ours)** $\alpha = 0$ | 32-bit, CPU | Polyhedra | S | S | U | U | S |

Table 5: The output interval on the $1 + 1 + ... + 1 + 2^{53} - 2^{53}$ (Order3 detector) pattern.

| **Algorithm** | **Bound** $[l, u]$ |
|---|---|
| IBP | $[512, 512]$ |
| IBP + Miné (2004) | $[500, 524]$ |
| FPSoundIBP | $[-1548, 2573]$ |

## C.1 SCALING THE PARAMETER $\Delta$

Here, we consider the sensitivity of our error parameter $\Delta$, and experiment with scaled versions $\alpha\Delta$, with $\alpha \in [0, 1]$. Recall that, when $\alpha < 1$, soundness is not currently guaranteed, our theoretical guarantees require $\alpha = 1$. Note that we have not proven that $\Delta$ is tight.

We first repeated the experiments in table 1 with $\alpha < 1$, the results are shown in table 4. We can observe that the verification result remains sound when $\alpha = 0.25$, for all the tested networks. However, for $\alpha \leq 0.2$, the result is incorrect for Order3, the verifier misses the backdoor.

However, one needs to be careful when interpreting these results. The smallest $\alpha$ that preserves correctness—which determines the practical extent of over-estimation—depends strongly on both *the specific numerical pattern of weights in the adversarial network and the order in which the interval sum is evaluated*. We illustrate this with the backdoor pattern of the Order3 network, where the sum is

$$\underbrace{1 + \cdots + 1}_{512\times} + \omega - \omega. \tag{18}$$

When evaluated in the default left-to-right order using interval arithmetic, the result is $[512, 512]$, meaning that no numerical errors arise in this particular summation order. However, the floating-point values that need to be bounded are in $[0, 512]$. In contrast, FPSoundIBP with $\alpha = 1$ yields the interval $[-1548, 2573]$, which introduces considerable over-estimation. Table 5 includes these output intervals, along with the one used by the (not practically sound) error bounding technique of Miné (2004), while table 6 shows the output ranges obtained for different values of the scaling parameter $\alpha$ with FPSoundIBP.

Let us now consider a different ordering of the sum, namely

$$\omega + \underbrace{1 + \cdots + 1}_{512\times} - \omega. \tag{19}$$

When evaluated with interval arithmetic (that is, $\alpha = 0$), the resulting interval is $[0, 1024]$, which correctly bounds the floating-point range $[0, 512]$. This means that $\alpha$ can safely be set to 0 for this expression tree while preserving soundness (IBP is already sound). In contrast, FPSoundIBP ($\alpha = 1$) yields the interval $[-2059, 3083]$—an interval that is wider than the one obtained with the original expression-tree. This example clearly demonstrates that the degree of over-estimation introduced by our method is sensitive to the expression tree. Also note that *the soundness itself is not sensitive to the expression tree: this is our main result*. But the width of the interval does depend on the particular expression tree.

Table 6: Lower and upper bounds obtained in the default order of the Order3 backdoor pattern by FPSoundIBP with various values of $\alpha$.

| Scale | Lower | Upper |
|---|---|---|
| 0.00 | 512.0 | 512.0 |
| 0.05 | 406.0 | 618.0 |
| 0.10 | 304.0 | 721.0 |
| 0.15 | 200.0 | 825.0 |
| 0.20 | 98.0 | 926.0 |
| **0.25** | **-6.0** | **1030.0** |
| 0.30 | -108.0 | 1133.0 |
| 0.35 | -210.0 | 1234.0 |
| 0.40 | -314.0 | 1338.0 |
| 0.45 | -416.0 | 1441.0 |
| 0.50 | -520.0 | 1545.0 |
| 0.55 | -621.0 | 1646.0 |
| 0.60 | -724.0 | 1749.0 |
| 0.65 | -828.0 | 1853.0 |
| 0.70 | -929.0 | 1954.0 |
| 0.75 | -1033.0 | 2058.0 |
| 0.80 | -1136.0 | 2161.0 |
| 0.85 | -1237.0 | 2262.0 |
| 0.90 | -1341.0 | 2366.0 |
| 0.95 | -1444.0 | 2469.0 |
| 1.00 | -1548.0 | 2573.0 |

Table 7: Comparison of the output interval lengths on sums of $n$ i.i.d. random numbers drawn uniformly from $[0, 1)$.

| n | IBP | FPSoundIBP | IBP + Miné (2004) |
|---|---|---|---|
| $10^1$ | $4.99 \cdot 10^{-16}$ | $3.67 \cdot 10^{-14}$ | $7.93 \cdot 10^{-15}$ |
| $10^2$ | $8.53 \cdot 10^{-14}$ | $3.63 \cdot 10^{-12}$ | $6.91 \cdot 10^{-13}$ |
| $10^3$ | $1.06 \cdot 10^{-12}$ | $3.49 \cdot 10^{-10}$ | $8.23 \cdot 10^{-12}$ |
| $10^4$ | $3.43 \cdot 10^{-11}$ | $3.58 \cdot 10^{-8}$ | $2.49 \cdot 10^{-10}$ |
| $10^5$ | $2.32 \cdot 10^{-9}$ | $3.57 \cdot 10^{-6}$ | $1.63 \cdot 10^{-8}$ |
| $10^6$ | $1.42 \cdot 10^{-7}$ | $3.55 \cdot 10^{-4}$ | $9.98 \cdot 10^{-7}$ |
| $10^7$ | $2.62 \cdot 10^{-6}$ | $3.54 \cdot 10^{-2}$ | $1.84 \cdot 10^{-5}$ |
| $10^8$ | $4.87 \cdot 10^{-5}$ | $3.54$ | $3.41 \cdot 10^{-4}$ |
| $10^9$ | $6.4 \cdot 10^{-3}$ | $354.328$ | $4.48 \cdot 10^{-2}$ |

### C.2 THE EFFECT OF THE NUMBER OF SUMMANDS

We generated $n$ i.i.d. random numbers uniformly distributed in $[0, 1)$ and compared the diameters of the resulting output intervals produced by no error bounding (IBP), the technique of Miné (2004), and our FPSoundIBP. The results are shown in table 7. It can be seen that the amount of *relative* over approximation also grows with $n$, a phenomenon predicted by theory, since $\Delta$ depends on $n$. Fortunately, in practice the fan-in of neurons is of a relatively moderate size.

## D ILLUSTRATIVE EXAMPLE FOR FPSOUNDIBP

Let us compute FPSoundIBP in full detail, step-by-step, for a simple input expression:

$$1 + 1 + 2^{53} - 2^{53}, \tag{20}$$

assuming a binary64 floating-point representation. In this representation, the true range of every possible expression tree for this sum is $[0, 2]$. For example, if we sum from left to right we get 2, and

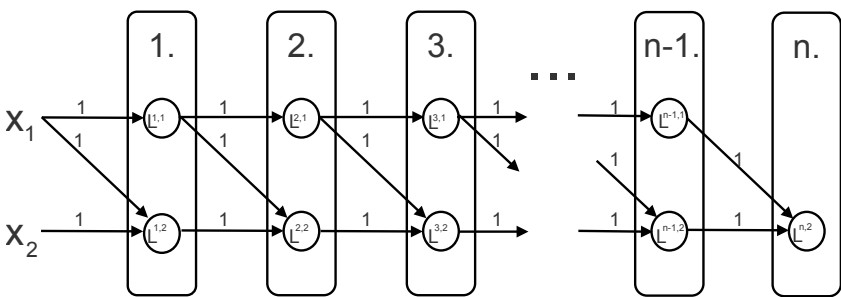

Figure 3: The structure of the example network to illustrate FPSoundSymbolic.

if we sum in the order of $1 + 2^{53} + 1 - 2^{53}$, we get zero. No values outside this range are possible (one can test every possible ordering).

**Verified expression tree.** First of all, FPSoundIBP works with an explicit expression tree, that we need to fix. We proved that the output interval will contain every possible expression tree, independently of which expression tree we chose here, nevertheless, the method needs a specific tree, and the actual output interval depends on this tree. Let this expression tree be represented by computing $((1 + 1) + 2^{53}) - 2^{53}$ from left to right.

**Plain IBP fails.** The IBP algorithm without our error bounding computes the interval $[2, 2]$ on this expression tree, because every operation in the tree (namely, $1 + 1$, $2 + 2^{53}$, and $(2 + 2^{53}) - 2^{53}$) can be performed without any rounding errors in floating-point, thus the result is a zero-length interval around the exact value. However, as we saw above, there is another expression tree which results in zero, and zero is not included in $[2, 2]$. Also note that if we used the order $1 + 2^{53} + 1 - 2^{53}$ then IBP would be sound (but only by chance).

**The computations of FPSoundIBP.** First, let us compute $\Delta$ according the its definition $\Delta = \frac{n\mu}{1 - n\mu}$ given in section 4.1. Here, $n = 4$ is the number of summands and $\mu = 2^{-52}$ is the machine epsilon, which gives

$$\Delta = \frac{4 \cdot 2^{-52}}{1 - 4 \cdot 2^{-52}} \approx 8.88 \cdot 10^{-16}. \tag{21}$$

Now, we can substitute into eq. (5) which gives

$$1[1 - \Delta, 1 + \Delta] + 1[1 - \Delta, 1 + \Delta] + 2^{53}[1 - \Delta, 1 + \Delta] - 2^{53}[1 - \Delta, 1 + \Delta] = [-18, 23]. \tag{22}$$

This interval over-approximates the true interval ($[0, 2]$), nonetheless, it is a sound over-approximation that is guaranteed to cover every possible expression tree.

Note that FPSoundIBP works with eq. (6) with interval inputs, but if the input contains scalars (that is, zero length intervals like $[1, 1]$) then this equation falls back to the inner product case that we used above.

# E    VERIFYING AN INTERESTING EXAMPLE SYMBOLICALLY

Here, we present an interesting neural network instance to demonstrate an issue with symbolic propagation methods. The issue is that—due to the extremely different expression trees that are used during verification and in deployment—the output of numerically and symbolically evaluated networks can drastically differ when using floating-point arithmetic. Our example network is designed to illustrates this, and we also explain how our symbolic error propagation method FPSoundSymbolic bounds both symbolic and numeric evaluations correctly for this example.

Consider a neural network with two inputs $x = (x_1, x_2)$, $n$ layers, and a single output. Let each hidden layer contain two neurons: $L^{i,1}$, and $L^{i,2}$, except for the final layer, which has only one neuron $L^{n,2}$. The upper neuron $L^{i,1}$ is connected to both neurons in the subsequent layer, while the lower neuron $L^{i,2}$ is connected only to the lower neuron of the next layer, $L^{i+1,2}$. All connection weights are fixed to 1, and all biases are set to 0. The structure is illustrated in fig. 3.

It is straightforward to observe that, due to the connectivity pattern and the choice of unit weights and zero biases, the contribution of $x_1$ accumulates linearly across the $n$ layers, while $x_2$ propagates directly to the output. Consequently, under symbolic evaluation, the network output is $x_2 + nx_1$.

However, when the network is evaluated using standard floating-point arithmetic, numerical errors may arise. For instance, if we choose $x_1 = 1$ and $x_2 = 2^{53}$ then, under the assumption that the sum $2^{53} + 1$ is rounded down to $2^{53}$ due to limited precision in IEEE 754 double-precision arithmetic (the contribution of $x_1$ is effectively lost at each layer). Consequently, the network will numerically evaluate to $2^{53}$, instead of the symbolic output, which would be close to $2^{53} + n$, with only a small possible deviation due to floating-point rounding. This means that *arbitrarily large discrepancies can arise between symbolic and floating-point evaluations*, growing with the number of layers $n$.

In double-precision floating-point arithmetic, the machine epsilon is given by $\mu = 2^{-52}$. In our network, there are two types of neurons: those with a single input and those with two inputs. Accordingly, we have two types of $\Delta$, denoted as $\Delta^{(1)}$ and $\Delta^{(2)}$. The intervals generated by these are denoted as $I_i = [1 - \Delta^{(i)}, 1 + \Delta^{(i)}]$, where $i = 1, 2$. Note that in the main text we used a common $\Delta$ for every layer (using the maximal number of inputs) for simplicity, however, one can use layer-wise $\Delta$ in this manner.

It can be seen that the extended symbolic expression for the $i$-th layer neurons are given by the following formulas:
$$L_\Delta^{i,1} = I_1^i x_1 + 0 x_2, \text{ where } i = 1, \ldots, n-1$$
$$L_\Delta^{i,2} = \left( \left( \ldots \left( (I_2^2 + I_1 I_2) I_2 + I_1^2 I_2 \right) \ldots \right) I_2 + I_1^{i-1} I_2 \right) x_1 + I_2^i x_2 \text{ where } i = 1, \ldots, n$$

Thus, the symbolic result for the output neuron is
$$L_\Delta^{n,2} = \left[ \left( \ldots \left( (I_2^2 + I_1 I_2) I_2 + I_1^2 I_2 \right) \ldots \right) I_2 + I_1^{n-1} I_2 \right] x_1 + I_2^n x_2$$

Let us look at the $L_\Delta^{n,2}$ interval formula and examine whether, for $x_1 = 1$ and $x_2 = 2^{53}$, it contains $2^{53}$. Since the exact coefficient of $x_1$ is $n$, we know that the interval coefficient of $x_1$ will include $n$, hence multiplying with $x_1$ will include $nx_1$. For the second part, the coefficient contains $[1 - n\mu, 1]$, because $\Delta^{(2)} > \mu$. Hence the $x_2$ term will include the $[x_2(1 - n\mu), x_2]$ interval. Since $\mu x_2 \geq x_1$, the final result will therefore also contain $x_2$.

