# OpenReview forum: "Sound Verification of Deployed Neural Networks"
_ICLR.cc/2026/Conference — Submitted to ICLR 2026_

### Official Review · Reviewer_aJmt · 2025-10-16

**Soundness:** 4
**Presentation:** 3
**Contribution:** 2
**Rating:** 8
**Confidence:** 4

**Summary:**

Many neural network verifiers operate on a theoretical model, ignoring many practical aspects such as floating point inaccuracies.
This paper addresses this by designing verification algorithms that are also sound under these practical aspects.

**Strengths:**

- Coming from a formal verification perspective that usually ignores these practical aspects,
it's good to see that the field also explicitly expands in this area.
- The paper is easy to read, making a good trade-off between technical description and providing intuitive understanding.
- All technical details are rigorously developed.

**Weaknesses:**

- The contribution could be seen as marginal, given that verification under floating-point precision has been researched before.
- A running example could help with the intermediate understanding
- Only two simple verification algorithms are shown, which trivially also suffer from such constraints (e.g., huge output intervals).
- Explicitly stating the notation instead of just referencing other papers makes the paper self-contained.
- The evaluation could be expanded, e.g., on different data sets.

**Questions:**

- I believe VNN-COMP'25 had some issues with tolerances regarding floating points. Do you think these would have been resolved if all tools used your approach?

---

> ### Author Response · Authors · 2025-11-22
> **response**
>
> Thank you for the positive feedback. Let us address the raised weaknesses and questions.
>
> **Contribution.** Indeed, the issue of floating-point representation has been discussed before in several contexts, mostly focusing on how to execute the verification algorithm itself so that it correctly bounds the theoretical output range of the neural network. However, in our case, we address a relatively new angle, namely the difference between deployed and theoretical models of computation. To the best of our knowledge, we are the first to propose a method that is provably sound in an environment that might select any valid expression tree to compute the neural network. Most importantly, we also provide a theoretical proof for the case of a symbolic propagation algorithm (Proposition 3), which is especially interesting because during symbolic propagation, the computed expression trees markedly differ from those in deployment environments, yet our bounding method is still correct, which we consider far from evident.
>
> **Running example.**  We now present the full derivation of a simple IBP example in the Appendix. Note that the original appendix already contained a derivation for the more challenging symbolic case, but the simpler IBP approach also benefits from an illustrative example that helps our readers understand the basic concepts and the approach.
>
> **Huge output intervals.** We are not sure what the reviewer intended to criticise when saying our algorithms suffer from huge output intervals, but maybe the fact that IBP in itself already produces significant over-approximation. True, but, for example, the symbolic approach we examine is much better in this regard in the case of benign (not malicious) networks. Also, we believe that our technique can be used as an add-on to existing methods, because whenever a sum is approximated, one can use our bounds to make it practically sound. We present only two simple examples to illustrate this methodology.
>
> If the reviewer meant that our bounding method introduces huge over-approximation relative to the baseline methods, that is not true for benign networks. We include new tables in the main text (Tables 2 and 3), as well as a new section in the Appendix on over-approximation (Section C), where some of these issues are discussed in more detail. For malicious networks, yes, the intervals explode, but that is the intended, correct behavior, because it indicates that a backdoor was found. Such an explosion is not expected for normal, benign networks.
>
> **Paper should be self-contained.** We agree that a paper should be self-contained as much as possible, and we made an effort to achieve this. At this point, we believe that all the concepts and notation that we rely on have been introduced. In the present version, several clarifications have been made, and the paper has undergone another proofreading round. We are more than happy to address any remaining presentation issues if they are pointed out to us.
>
> **Expand evaluation.** We expanded our evaluation by providing more experiments regarding over-approximation that we include in the Appendix in Section C. However, our experimental results play a secondary role in this work, and they are either for the validation of our theoretical results or for the purpose of making a few qualitative points on how the method behaves when a malicious backdoor is added, and when the network is benign.  It would be interesting to conduct a full-scale quantitative study as part of our future work.
>
> **VNN-COMP'25.** The specification of the task was not well-defined. If the task had been the verification of the theoretical (full-precision) model, then some existing methods could have provided real soundness in this sense (e.g., DeepZ), and our method would not have been necessary. If the task had been to verify the deployed network, then the task specification should have included every detail of the deployment environment. If this environment does not involve approximate algorithms for computing the network (for example, approximate matrix multiplication), and no under- or overflow occurs, then yes, our method guarantees soundness (and we are not aware of other algorithms that do).

---

> > ### Comment · Reviewer_aJmt · 2025-11-23
> >
> > Dear Authors,
> >
> > Thank you for your comments. My concerns are adequately addressed and I will keep my already positive score.
> >
> > Best, Reviewer aJmt

---

### Official Review · Reviewer_y7Wb · 2025-10-23

**Soundness:** 4
**Presentation:** 2
**Contribution:** 4
**Rating:** 6
**Confidence:** 5

**Summary:**

Existing neural network verifiers assume real arithmetic. Some account for floating-point error (FP error), but none of the existing verifiers account for the non-deterministic execution order of floating-point operations in modern architectures. This paper describes how existing bound propagation techniques for neural network verification can be extended to be sound under FP error and non-deterministic execution orders. The approach is based on bounding the maximal FP error that can accumulate from different execution orders for the primitive operations of a linear operation. Growing the bounds computed by interval arithmetic and zonotope propagation for neural networks by this FP error bound and using outward rounding makes these bound propagation approaches sound under FP error and non-deterministic execution orders. The experimental evaluation demonstrates that this additional rounding can detect several attacks on verifiers from the literature. The runtime overhead of the extension is up to 26% compared to an implementation without FP soundness.

**Strengths:**

This paper is a step towards extending the certificates provided by neural network verifiers to the real execution environments in which neural networks are deployed. I am not aware of comparable approaches in the literature. The proofs are correct, and the experimental evaluation is convincing.

**Weaknesses:**

The only significant weakness of this paper is its sloppy presentation. Please refer to the list of presentation issues below. I will raise my rating to "accept" if the authors address these issues during the rebuttal window.

Beyond this, a weakness of this paper is that it can not address additional peculiarities of the hardware on which neural networks are deployed, so that the guarantees provided by this work still remain unsound in practice, as mentioned in the limitations section. Another weakness is that the paper only provides explicit methods for intervals and zonotopes, not for the most widely used polytope relaxations. In my opinion, both weaknesses are insignificant, given that research on extending the guarantees of neural network verifiers to real deployment environments is extremely sparse. Lastly, a hypothetical limitation of this paper is that its analysis breaks down if the product of the hidden layer width and the machine precision exceeds one, which might happen for very large networks in extremely quantised execution environments.

#### **Presentation Issues**
1. Reading the abstract, I did not know what "expression tree" referred to. Talking about the order of primitive operations, such as addition and scalar multiplication, would be easier to understand.
2. Instead of talking about "reasonable assumptions" in line 52, state that you provide soundness for non-deterministic execution orders, but not, for example, special GPU algorithms for matrix multiplication.
3. Quantify the "reasonably low overhead" in line 66. Also report the absolute numbers alongside the percentages.
4. Put brackets around your citations when they are not part of the sentence. For example, "is known to be NP-complete (Katz et al, 2017)" in line 75.
5. The sentence "sound (but not *necessarily* complete) verifiers aim to ... at the expense of completeness" is contradictory. You are talking about sound incomplete verifiers.
6. Please settle on one term for linear bound propagation instead of referring to it as "symbolic", "linear" (line 80), and "Polyhedra" (Table 1).
7. I appreciate that you cite early references for many approaches, such as Miné (2004) for linear interval expressions. In this spirit, there are clearly earlier references for IBP than Xu et al. (2020).
8. Stating that the outcome of an associative operation can strongly depend on the execution order in line 107 is contradictory.
9. The norm is not $p$ but ${\| \cdot \|}_p$ in line 127.
10. Your definition of $P(x^\ast)$ states that the *value* of the largest output of the network for $x$ needs to match the *value* of the true-class output for the reference input. That does not make sense. Since you have already introduced the notation, it is much easier to state $P(x^\ast) = \{x : y(x^\ast) = y(x)\}$. There are similar slip-ups in the remainder of the text.
11. omit the "very" in "very different" in line 161 or write "markedly different".
12. The $D_{p, \epsilon, \ldots}$ in line 177 misses a $(x^\ast)$.
13. "binary operation" is an ambiguous expression in line 186. I understood it to mean "expression with two arguments". However, IBP also uses intervals for unary operations. I think "binary" can safely be omitted here and in the remainder of the paper.
14. The sentence in line 190 is unclear. Besides, it could probably use a citation to "On the Effectiveness of Interval Bound Propagation for Training Verifiably Robust Models" by Gowal et al.
15. Introduce the $\delta_i$ variables more clearly. In particular, $\delta_i \in \mathbb{R}$. At first, I thought $\delta_i$ was a positive constant.
16. Similarly, introduce $\delta(n)$ in Lemma 1.
17. I think $l, u$ should lie in $X$, not $\mathbb{R}^n$ in Proposition 2.
18. Avoid formulations like "it is obvious" in line 344. Instead, give a brief reasoning.
19. Your IBP ReLU relaxation is faulty. It should be $\max(l, 0)$, not $\min(l, 0)$ in line 248.
20. By "affine arithmetic" in line 362, are you referring to real arithmetic as opposed to floating-point arithmetic?
21. Line 368, "Of course, more sophisticated approximations can also be used". If you have an FP-sound formulation of CROWN, please provide it. Otherwise, strike this sentence.
22. What is $h$ in line 413?=
23. Also provide absolute runtime overheads in sections in line 445.
24. The references contain duplicated URLs and DOIs.
25. Some DOIs, such as 10.1007/978-3-319-77935-5\_9 are unresolvable.
26. Some abbreviations and tool names are not capitalized correctly in the references, such as "Intervalarithmetic.jl".

I did not read the entire appendix, but please also correct issues similar to the above in the appendix.

**Questions:**

- Equation (9) requires a `np.where` operation, or similar, to select among the three cases in a vectorised computation. Can that lead to any numerical issues?

---

> ### Author Response · Authors · 2025-11-22
> **Response**
>
> First of all, we thank the reviewer for the exceptionally thorough proofreading, which we found to be a valuable resource for improving the presentation. Let us reply to some of the points that were raised.
>
> ## Weaknesses and questions
>
> **Focus on non-associativity.** Indeed, we address only the problems that directly stem from the non-associativity of floating-point arithmetic, but there are other sources of errors, as we acknowledge among the limitations. Nevertheless, as the reviewer agrees, dealing with non-associativity is an important step towards practical robustness.
>
> **Only linear interval expressions.** Indeed, we discuss only linear interval expressions for symbolic propagation. However, note that this case required a significantly deeper analysis and a more complex proof than that of IBP, despite the formulation of Proposition 3 looking similar to the IBP result in Proposition 2. We found this connection especially interesting and somewhat unexpected, which could lead to further generalizations.
>
> **Very large networks and large machine epsilon.** If the machine epsilon is very large (low resolution floating point), then we cannot deal with large sums, for example, in the IEEE binary16 floating-point format, the maximum is $n=1023$ summands. However, for IEEE binary32, n is already approximately  $8.39\cdot 10^6$, which is more than enough in practice. Also note that $n$ is not the size of the entire network but the fan-in of any neuron, which is reasonable even in very large networks. For quantized networks, clearly, different approaches of verification are necessary, but such networks are often exact even in deployment and thus do not suffer from the problem we are tackling.
>
> **Numerical behavior of eq (9).** We introduce no numerical issues in equation (9) because $[l,u]$ is computed safely using concretization with over-approximation, and subsequently, we test only for positivity and non-positivity, which are precise operations even in floating-point, so we do not lose soundness.
>
> ## Presentation suggestions
>
> We carefully considered all 26 suggestions. Here, we address only those that were not completely evident and straightforward or that need explanation. The ones we do not address were simply implemented.
>
> 1: The term “expression tree” was removed from the abstract, and we made sure it is defined before using it, which we keep doing because expression trees are a key concept in our setup.
>
> 3: We now report the relative overhead when listing our contributions, but we report absolute numbers only in Section 5 (as per your comment no. 23), because to interpret those, one needs to understand the details of the example verification problem we tackle as well as our implementation, which cannot be given in advance. Although the relative errors also depend on these, they are still much better indicators of the cost of our method in the absence of the necessary context.
>
> 6: We reformulated our discussion and made it clearer that linear expressions (now first mentioned only in section 3.4) are only one way of implementing symbolic propagation, and that in this paper, we use only linear expressions for symbolic propagation. From the table, we removed the terms Polyhedra and Zonotope and replaced them with Symbolic because, indeed, these concepts were not introduced previously and are not needed for understanding our results.
>
> 7: We included two new references: (Alefeld & Herzberger (1983); Gowal et al. (2018)).
>
> 17: Although $[l,u]$ represents the input, for which $l,u \in X$ in deployment, Proposition 2 is true for $l,u \in \mathbb R$, so it is stated in this more general form.
>
> 20: Affine arithmetic is a technique for symbolic interval propagation that represents intervals as affine functions of range variables. We simply removed the confusing reference to affine arithmetic, as it is not necessary to follow the discussion (that is, it is evident that we do not use affine arithmetic here, so there is no need to state it).
>
> 25: We checked all the URLs, and each of them worked when we tested them. The given DOI had a typo, but we have now removed the DOI and only kept the (working) DOI URL.

---

> > ### Comment · Reviewer_y7Wb · 2025-11-23
> >
> > Thank you for revising your paper and your explanations for my questions. I updated my recommendation and have no further questions.

---

### Official Review · Reviewer_rzs3 · 2025-10-30

**Soundness:** 1
**Presentation:** 2
**Contribution:** 3
**Rating:** 4
**Confidence:** 3

**Summary:**

Formal verification of neural networks does not guarantee safety in deployed models due to technical factors affecting floating-point computations, such as precision and computation order. The paper follows a method [Higham, (2002)] to bound the backward error of inner products and introduces a technique to formally bound the output value of a computation regardless of its execution order. In addition, the authors propose two improved versions of existing bounding methods (FPSoundIBP and FPSoundSymbolic). These methods are compared to the original versions (SoundIBP and SoundSymbolic) in terms of soundness, runtime, and accuracy (output-range similarity). The results demonstrate that the new versions guarantee soundness while preserving similar runtime.

**Strengths:**

1. The paper addresses relevant challenges in the formal verification of neural networks.
2. It takes a first step toward the formal verification of deployed neural networks.
3. The method is integrated into two common bounding mechanisms used in formal verification.

**Weaknesses:**

1. Limited evaluation: rows 1-10 in Table 1 appear in previous work [Szász et al. (2025)], so the new experimental results include only rows 11-12 in Table 1, which only confirm the theoretical results but do not supply additional information. Figure 1 and Table 2 are based on experiments with 100 input samples and one model trained on MNIST.

2. Limited soundness:
- The claim that the method preserves runtime is not supported by the results for IBP, where the runtime increased by ~25%.
- The claim (Line 27, Line 65) that the method preserves accuracy (output-range) is not supported in the only check with respect to Order3 attack, where the output range is much larger for Order3. The method should be compared in other environments (Pr., Order1, Order2, Zombori et al. (2021)) as well.

3. Scalability: The assumption that $n\cdot\mu<1$ limits the scalability of the proposed method (if $n>1/\mu$). Moreover, multiplying $(2n − 1)$ times by $\Delta$ seems to significantly increase the bounds on the result.

4. Missing related literature: No prior work on formal verification of quantized networks [1, 2, 3] is mentioned.

5. Readability issues:

- The term “expression tree” appears five times (two of them in the abstract) before being explained at Line 215.

- The authors claim they are “closely following [Higham (2002)]” to bound inner products, but the technique of [Higham (2002)] is not explained at all, although it appears to be the core of the proposed bounding method.

- Line 361: “similar to DeepZ but without using affine arithmetic.” DeepZ's details are not explained in the paper.

- Line 413: What is h?

A. Towards Efficient Verification of Quantized Neural Networks (Huang et al., AAAI 2024).

B. QVIP: An ILP-based Formal Verification Approach for Quantized Neural Networks (Zhang et al., ASE 2022).

C. Scalable verification of quantized neural networks (Henzinger et al., AAAI 2021).
QVIP: An ILP-based Formal Verification Approach for Quantized Neural Networks (Zhang et al., ASE 2022)
Scalable verification of quantized neural networks (Henzinger et al., AAAI 2021).

**Questions:**

1. The work does not support complete verification. It is recommended to change the title to a more modest one (e.g., “Towards…”).

2. It is stated (Lines 240-243) that the error of each calculation can be expressed as a multiplication by
$(1+\delta_i)$, but there is an example (Lines 153–155) where the result changes from 0 to 1. How does multiplication correct the error in this case?

3. Can the authors share the (average) values of $n$, $\delta_i$ and $\Delta$​ in the experiments? What is the average ratio $\delta_i/\Delta$? It can help the reader to approximate the effect on the output range.

4. It seems that all rows in Table 1, except the last two, are taken from [Szász et al. (2025)]. Is that correct? Why didn’t the authors state this explicitly?

---

> ### Author Response · Authors · 2025-11-22
> **response**
>
> Thank you for the detailed feedback and the suggestions. Let us address the comments and questions below, mentioning the changes made to the paper when applicable.
>
> ## Addressing weaknesses and recommendations
>
> **Scalability.** Indeed, if the machine epsilon is very large (coarse-grained quantization), then we cannot deal with large sums, for example, in the IEEE  binary16 floating-point format, the maximum is $n=1023$ summands. However, for IEEE binary32, n is already approximately $ 8.39\cdot 10^6$, which is more than enough in practice. As for adding $\Delta$ $2n-1$ times: in fact, we add it only $n$ times, but yes, the over-approximation of a single inner product grows with $n$ in the worst case. We now include a new section in the appendix on over-approximation (Section C), which shows that it also depends on the actual sum we examine, as well as the size of the sum.
>
> **Presentation issues.**
> - The term “expression tree” is now properly introduced.
> - We removed the reference to $h$ altogether.
> - In fact, we do include every detail we use from (Higham, 2002) in Section 4.1.  Note that our main contribution is not this technique, but its application to both IBP and symbolic verification methods, along with the theoretical proofs of their soundness.
> - From DeepZ we use only the main idea of the ReLU envelop definition, which is given in full in the paper.
>
> **Quantized networks.** We now mention quantized networks in the related work section.
>
> **Limited empirical evaluation.** Our contributions are mainly theoretical. The experimental results are indeed only for validation and for presenting qualitative illustrations of some interesting behaviors. The soundness results in Table 1, in particular, were indeed expected and predicted by theory. We now make it clearer that Table 1 contains rows from related work. Our results on overhead are also mainly qualitative and not quantitative. We wanted to illustrate a few phenomena, like the effect of including backdoors into the verified network.
>
> In the current version, we added a new section in the Appendix about over-approximation (section C), where we include results on the sensitivity of the $\Delta$ parameter as well as the overhead of the method on random sums of different sizes. We also expanded the set of results in the main paper, and we now present the predicted width of baseline methods as well for comparison in Tables 2 and 3.
>
> **Limited soundness.** A 25% increase in running time is a runtime of the same order of magnitude, which is highly non-trivial in the area of verification, and we still consider it a small price. As for the output range, the output interval length does remain small for normal (not malicious) networks; we include more data on this in the new version, including more malicious networks (Order1 and Order2) as requested, in Tables 2 and 3. The extreme increase in length that we see for some malicious networks is, in fact, desirable, because that indicates that the backdoor was caught. We explain this in a separate paragraph (wide output is a red flag). We also include detailed measurements on over-approximation in the appendix in Section C, which also supports that substantial over-approximation happens in benign networks only for extremely large sums well beyond practical instances.
>
> ## Answers to the questions
>
> **Title.** We would be happy to prefix it with “Towards”; however, the openreview form does not allow changing the title at this point. If there is a possibility to change the title consistently later, we promise to do that.
>
> **How does the multiplication correct the error?** We include a detailed derivation using a simple example that illustrates this question in the Appendix in Section D. In the specific example you cite, in the first order, $\omega+1$ is not zero theoretically, so it will be multiplied by $1-1/(1+\omega)$ (note that this means $\delta_1=-1/(1+\omega)$) to reconstruct $\omega$, the floating-point result. The second (corrected) addition will be $\omega-\omega$, which is zero, so it does not have to be corrected, because it is also theoretically zero (so $\delta_2=0$). Every expression tree has a similar set of $\delta_i$, and we use the results of Higham (2002) to bound the overall effect of these $\delta_i$ corrections independently of which expression tree is used.
>
> **Values of parameters in experiments?** As for $n$, we now include a separate experiment to study its effect in the appendix in Section C.2. As for $\delta_i$, they are smaller than the machine precision $\mu$. We never explicitly compute them; they are just a theoretical stepping stone to compute the correct bound $\Delta$, as described in Section 4.1. The actual value of $\Delta$ can also be computed exactly using the machine precision $\mu$ and $n$. We, however, included an illustrative example in the Appendix in Section D, where every parameter is explicitly calculated.
>
> **Table 1.** Please see above.

---

### Official Review · Reviewer_7KQz · 2025-11-01

**Soundness:** 3
**Presentation:** 1
**Contribution:** 3
**Rating:** 6
**Confidence:** 3

**Summary:**

Existing verifiers typically validate only the ideal mathematical model of a neural network, whereas real-world deployments can introduce deviations due to floating-point precision, operation ordering, and parallel execution. Attackers may exploit these discrepancies to embed deployment-specific backdoors: a model appears safe during verification but behaves maliciously once deployed. This paper aims to incorporate all possible numerical execution behaviors that may occur during deployment into the verification process to ensure that robustness guarantees hold in realistic execution environments. For example, after a model is trained, an attacker may target a specific deployment platform with certain numerical properties, identify neurons whose behavior is highly sensitive to these properties, and craft trigger inputs that activate them only on that platform, thereby manipulating the model’s output while bypassing verification. To defend against such threats, the authors derive a order-independent relative error bound ∆ based on backward floating-point error analysis and use it to widen the inner-product computations so that the resulting intervals provably cover all possible floating-point outcomes across deployment environments. This ensures that verification conclusions remain valid under any feasible deployment execution.

**Strengths:**

1. The paper is among the first to explicitly address minute numerical discrepancies across deployment scenarios and demonstrate that these can be exploited to create stealthy backdoors affecting model trustworthiness.
2. The mathematical formulation and soundness proof are rigorous and professionally presented.
3. The structure and argumentation are generally well organized.

**Weaknesses:**

1. The motivation and threat model remain abstract. The description of deployment-specific backdoors in the Introduction is theoretical and may be difficult to follow for non-experts. A visual attack-flow illustration would significantly enhance clarity (e.g., attacker characterizing deployment → constructing environment-sensitive detector neuron → crafting trigger inputs → activation upon deployment).
2. Lack of flexibility and ablation for the widening parameter ∆. Although ∆ is theoretically derived, the paper does not evaluate the sensitivity of the verifier to its scaling (strictness vs. conservativeness trade-off). Ablation using a scaling factor (e.g., αΔ) or analysis across different depths / fan-ins would strengthen empirical understanding.
3. Insufficient explanation of runtime overhead causes. While empirical runtime curves are provided, the paper does not clearly articulate where the additional cost comes from (e.g., more unstable ReLUs → more linear relaxations → increased concretization). A short explanation—possibly with breakdown—would help guide future optimization.
4. Assumptions and applicability need clearer visibility. Some limitations (e.g., not addressing overflow/underflow, numerically approximated operations) appear only near the conclusion. These constraints are important in practice and should be highlighted earlier, including discussion of potential extensions.

**Questions:**

1. Could you include one or two illustrative diagrams in the Introduction showing:
(a) how an attacker detects or infers a target deployment setup,
(b) how environment-sensitive detector neurons are constructed or identified, and
(c) how trigger inputs activate malicious behavior only in deployment?
Using Order3 or the precision-based attack as an example would be highly instructive.

2. While ∆ is theoretically determined, could you comment on or experiment with its tunability?
For example, sweeping a scaling factor αΔ and reporting how robustness success rate, interval width, and runtime change? Also, how sensitive is ∆ to layer-wise fan-in? Are some layers more influential than others?

3. Could you provide a brief explanation or breakdown of runtime overhead sources?
Even a coarse analysis—such as contributions from interval scaling & outward rounding, unstable-ReLU relaxations, symbolic expression growth—would help clarify operational trade-offs and guide future improvements.

---

> ### Author Response · Authors · 2025-11-22
> **response**
>
> Thank you for your constructive comments and your encouragement. Here, we address the points raised.
>
> **Scaling the $\Delta$ parameter.** This question is very interesting as it directly asks for some measurements of the amount of over-approximation generated by our method. We included a new section in the appendix (Section C) where we present scaling experiments. However, we emphasize in this section that one has to be very careful when interpreting such experimental evidence, because the amount of over-approximation induced by the theoretical $\Delta$ very strongly depends on the order of the summands in which the verifier processes them. While our main result is that our computed interval is guaranteed to contain the true range for every ordering (in fact, every expression tree), the ordering does influence the amount of over-approximation.
>
> **Short explanation of runtime overhead.** We added a short explanation as the first paragraph of Section 5.2.1 on runtime overhead.
>
> **Visibility of limitations.** We now include the limitations explicitly in the introduction. There is an “assumptions” paragraph right at the beginning of Section 4 that we extended by mentioning under- and overflow as well. As for mitigations, these are non-trivial because the presence of approximate algorithms for arithmetic operations like matrix multiplication makes verification substantially harder. In this case, we need to integrate the approximate algorithm into our model that we verify, and, obviously, this also requires exact knowledge of what approximation happens for which neurons. We included more discussion in the limitations section on mitigating under- and overflow. In short, for IBP it is not hard, but for Symbolic it seems to be a more serious challenge.
>
> **Diagram about attacks.** We now include a rather detailed discussion on this issue in the appendix in Section B.1 entitled “Motivation and Attack Scenarios”. We hope a textual description is equally helpful in improving the understanding. However, we stress that the idea of such attacks is not the original contribution of this work; hence, we’d rather not put an extra emphasis on this in the main paper. We use this idea, along with the actual attacks, from related work, and the focus of the paper is on the novel theoretical contributions.

---

### Author Response · Authors · 2025-11-22
**common reply**

We thank all the reviewers for their thorough and constructive comments. We give detailed individual responses to all reviews; here, we provide an overview and summarize the main themes.

We revised the presentation of the paper based on the numerous useful recommendations and our own reading. The main paper now includes additional experimental results, a more detailed discussion, clarified references to related work, and an improved flow. The appendix contains a new section on new experimental results on the extent of over-approximation, and we include an illustrative example for IBP as well.

**Main contribution.** We wish to emphasize here that our contribution is mainly theoretical. Our results build on the error bounding method of (Higham, 2002), applied to two approaches to verification: interval bound propagation (IBP) and symbolic propagation using linear interval expressions (Symbolic). These are two key components that many verifiers apply in one form or another. For both, we prove theoretically that our method is sound in every possible expression tree. Most importantly, for the Symbolic approach, this result is far from evident because the expression trees used during verification never occur in deployment normally, so the proof of the soundness in this case is markedly harder. *This is the result we are most proud of.*

We also include experimental results, but the role of those is secondary, mainly to validate the theoretical results, and also to illustrate a number of qualitative properties of the method, in connection with applying it to malicious network instances. Nonetheless, these results were extended and now provide more insight into the extent of over-approximation.

---

### Meta-Review · Area_Chair_sUwS · 2026-01-14

**Summary:**

The paper proposes an error-bounding technique to make neural network verifiers sound when considering floating-point arithmetic and non-deterministic execution orders in deployment environments. Building on Higham (2002)'s backward error analysis, the authors develop methods to bound inner products such that the resulting intervals provably cover all possible floating-point outcomes across different expression trees. They demonstrate this approach through two algorithms, which are extensions to interval and symbolic bound propagation methods. The paper makes a valuable contribution by addressing a genuine gap in neural network verification, namely that existing verifiers only guarantee soundness for theoretical models but not deployed networks where floating-point operations can exhibit non-associative behaviors.

The AC disagrees with the emphasis that the paper is a “theoretical” paper. The theoretical contribution is quite weak compared to other ML theory papers accepted at similar conferences, as the key theorem on bounding dot product is a result of Higham (2002) and the extension to IBP and symbolic bounds are relative straightforward (since they use dot product as the elementary operation). On the other hand, the AC does believe this paper can bring great practical impact. The sound bound propagation methods, if implemented in practical verification tools, can address one key weakness in existing tools and provide additional confidence for verification results. Unfortunately, the paper’s empirical evaluation is extremely weak compared to most existing work published at this conference, such as Mn-BaB (Ferrari et al., ICLR 2022) and (De Palma et al., ICLR 2021). All experiments were done on a very small, single MNIST network with only 100 instances tested, and it is unclear whether the approach can effectively scale to and extend to any practical problems, such as these in VNN-COMP benchmarks.

The AC encourages the authors to continue working in this direction and make the work more solid. Since the algorithms are relatively straightforward, integrating it into one of the existing tools that can handle at least some larger and more practical benchmarks and will greatly improve the impact of this work. Especially, there is little incentive for existing tool builders to implement these algorithms because most neural network verification tools are built by academia, and students prioritize publishing papers rather than implementing existing algorithms. If no one implements these algorithms and enables their usage in practical scenarios, the impact of this work will be severely limited.

**Reviewer Concerns:**

The rebuttal and revision addressed several presentational and experimental concerns, expanded the discussion of limitations and assumptions in the paper. However, the limited scope of empirical evaluation persists as a fundamental weakness, while the authors added some experiments to the appendix, the core evaluation remains confined to MNIST with minimal new experimental validation beyond confirming theoretical predictions. Additionally, while limitations are acknowledged (e.g., approximate arithmetic kernels and under/overflow handling, especially for symbolic), these limitations are actually severe in almost all practical verifiers used today (e.g., using complex compute primitives beyond simple dot product).

**Reviewer Scores:**

Reviewer rzs3, who gave the most critical score of 4, along with a soundness rating of 1/poor, did not respond after the rebuttal. While the author's response addressed some technical details about how the error correction works and added experiments on over-approximation, the fundamental issues of limited empirical validation and scalability concerns were not resolved. The AC shares the same opinion as this reviewer and believes this work needs improvements for publishing at a top-tier ML conference.

---

### Decision · Program_Chairs · 2026-01-26

Reject